# SOLAR: COMMUNICATION-EFFICIENT MODEL ADAPTATION VIA SUBSPACE-ORIENTED REPARAMETRIZATION

## ABSTRACT

Parameter-efficient fine-tuning (PEFT) methods, such as LoRA, enable scalable adaptation of foundation models by injecting low-rank adapters. However, their communication and storage costs remain a major bottleneck in resource-constrained settings. We propose **SOLAR** (Subspace-Oriented Latent Adapter Reparameterization), a post-training compression framework that substantially reduces the communication cost (i.e., the number of parameters to transmit or store) of PEFT adapters. SOLAR expresses each PEFT update as a linear combination of basis vectors formed from the foundation model's singular vectors with controlled random perturbations. By exploiting the subspace similarity (the alignment of principal directions) between the foundation model and task-specific fine-tuned updates, SOLAR decouples the adapter size from PEFT structure and ensures compact yet expressive representations. It is model-agnostic and compatible with existing PEFT methods, including LoRA, AdaLoRA, and other adapter modules. We theoretically establish a bound on the reconstruction error. Experiments on language and vision tasks using LLaMA, GPT, and ViT models demonstrate that SOLAR preserves task performance while significantly reducing model representation sizes, offering an effective and communication-efficient solution for deployment in distributed systems and edge devices.

## 1 INTRODUCTION

Foundation models—large-scale pretrained transformer architectures—have catalyzed substantial progress across natural language processing, computer vision, and a range of other domains. However, adapting these models to downstream tasks remains resource-intensive. Full fine-tuning, which updates all model parameters, demands considerable computational, memory, and storage resources Houlsby et al. (2019). Parameter-Efficient Fine-Tuning (PEFT) techniques address this challenge by freezing the backbone and updating only a small set of task-specific parameters. For example, adapter modules insert compact trainable layers into each network block Houlsby et al. (2019); prefix-tuning optimizes a continuous prompt of only ∼0.1% of the model's parameters Li & Liang (2021); and Low-Rank Adaptation (LoRA) injects low-rank update matrices into each layer Hu et al. (2021). These methods achieve performance comparable to fully fine-tuned models while updating less than 1% of the model's parameters.

Despite these parameter savings, the cumulative communication and storage costs of PEFT modules remain a critical bottleneck in many real-world scenarios, particularly as foundation models continue to scale Wolf et al. (2020). In distributed scenarios (e.g., federated learning), these adapters must be communicated and stored across multiple devices or nodes, leading to significant overhead Wolf et al. (2020). Communication and storage overhead increase with the number of PEFT modules, as many fine-tuned adapters are saved and frequently transmitted or synchronized, thus turning millions of adapter parameters into a major bottleneck, particularly in bandwidth-limited or memory-constrained environments such as edge devices or federated learning systems Gao & Zhang (2024); Wang et al. (2025). The resulting communication and storage costs (i.e., the number of adapter parameters that must be transmitted and stored) can lead to slower training, increased energy consumption, and reduced scalability, highlighting the need for more efficient adapter compression techniques.

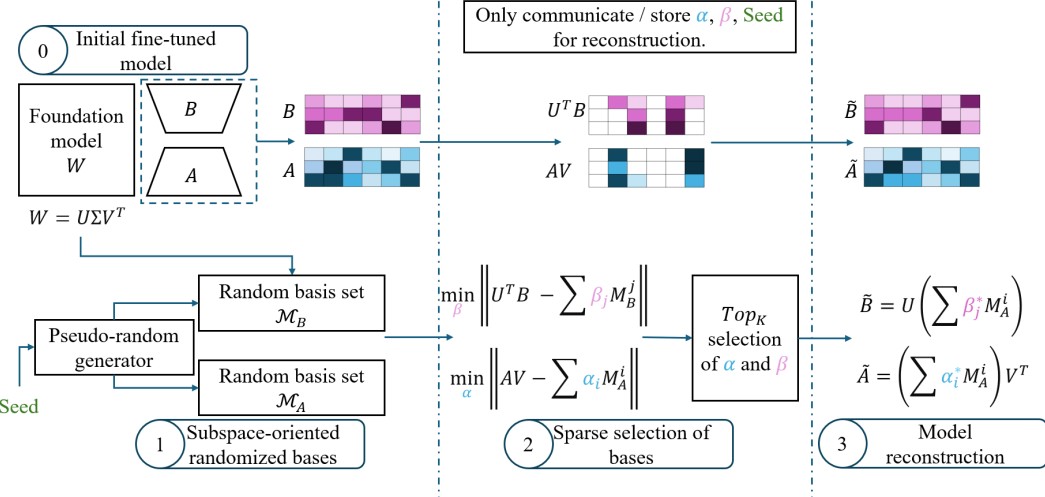

Figure 1: Overview of SOLAR. Given fine-tuned adapters $(A, B)$, SOLAR projects them onto structured subspaces derived from the pretrained model's SVD. A seeded pseudo-random generator (seeded with a known value) deterministically creates the basis matrices. Top-$k$ coefficients $\alpha$ and $\beta$ are selected under a budget to reconstruct $\tilde{A}$ and $\tilde{B}$, while the bases are never stored or transmitted. Only the coefficients $\alpha$, $\beta$, and the seed need to be communicated or stored.

To address this, several methods decouple tunable parameters from adapter rank and model dimensions: NOLA Koohpayegani et al. (2024) expresses LoRA's matrices as linear combinations of random basis matrices, training only the coefficients; VeRA Kopiczko et al. (2023) uses shared frozen random vectors with small learned scaling vectors; and SVFT Lingam et al. (2024) constructs a basis from singular vectors of pretrained weights and learns a sparse combination during fine-tuning. However, random bases not aligned with the model or task may reduce representational efficiency, and methods such as Kopiczko et al. (2023); Lingam et al. (2024); Koohpayegani et al. (2024) are not post-hoc, as they modify the training process and cannot compress adapters already trained—creating a need for a flexible, training-free compression utility.

In this paper, we propose SOLAR (**S**ubspace-**O**riented **L**atent **A**dapter **R**eparameterization), a novel post-training compression method for PEFT adapters. SOLAR exploits the empirical structure of adapter updates by reparameterizing them as linear combinations of structured, randomized basis matrices. It is model-agnostic and applicable post-training without modifying the fine-tuning process. The main contributions of this work are as follows:

- We leverage the observed subspace similarity between the foundation model's weights ($W$) and the task-specific update ($\Delta W$) to create a more compact and efficient adapter representation. By expressing $\Delta W$ as a sparse combination of basis vectors, our method effectively decouples the adapter's final size from the model's architecture.
- We develop a three-step framework for post-hoc adapter compression that involves: 1) constructing a basis pool of size $N$ by perturbing the foundation model's singular vectors with random noise, 2) performing a sparse selection of the most significant basis vectors to meet a budget $k$, and 3) reconstructing the adapter using only the selected coefficients and a single random seed.
- We provide a formal theoretical analysis that bounds the reconstruction error. Our proof decomposes the total error into the original training error and a controllable compression error, which can be minimized by tuning SOLAR's hyperparameters ($N$ and $k$).
- We demonstrate through extensive experiments that SOLAR reduces adapter sizes by up to 98% while preserving the performance of the original LoRA adapters. Our results show competitive accuracy across a wide range of vision and language tasks using ViT, GPT-2, and LLaMA models.

## 2 PROPOSED METHOD: SOLAR

We propose a *post-training* compression strategy that serves as a modular add-on for compressing PEFT-based updates. It introduces no training overhead and is compatible with LoRA Hu et al. (2021),

QLoRA Dettmers et al. (2023), Compacter Karimi Mahabadi et al. (2021), and NOLA Koohpayegani et al. (2024), operating post-hoc by taking the final trained adapter matrices as input. SOLAR applies to OFT Qiu et al. (2023) and variants Liu et al. (2023), compressing $\Delta W = (R - I)W$ via its SVD-based subspace without altering the orthogonal parameterization. By exploiting the low-rank structure of updates, SOLAR significantly reduces communication and storage costs in distributed or resource-limited settings.

## 2.1 PROBLEM FORMULATION

Transformer-based models parameterize attention and MLP layers using full-rank weight matrices $W \in \mathbb{R}^{m \times n}$. Recent PEFT methods, such as LoRA Hu et al. (2021), decompose the task-specific update $\Delta W$ as $\Delta W = BA$, where $A \in \mathbb{R}^{r \times n}$, $B \in \mathbb{R}^{m \times r}$, and $r \ll \min(m, n)$. This reduces the trainable parameters from $mn$ to $r(m + n)$, yielding a compression ratio of $\frac{mn}{r(m+n)}$. While effective, LoRA's fixed-rank formulation limits its flexibility. Alternatives, such as NOLA Koohpayegani et al. (2024), leverage random projections to approximate $\Delta W$, but often require large basis sets to sufficiently capture the relevant directions. To address this challenge and enhance compression further, we formulate the problem as minimizing the approximation loss between $\Delta W$ and its compressed counterpart $\Delta \tilde{W}$ subject to a strict communication (or storage) budget:

$$\min_{\Delta \tilde{W}} \|\Delta W - \Delta \tilde{W}\|_F^2, \quad \text{s.t.} \ \|\Delta \tilde{W}\|_0 \leq k, \tag{1}$$

where $\|\cdot\|_F$ denotes the Frobenius norm, and $\|\cdot\|_0$ represents the number of non-zero elements (i.e., $\|X\|_0 \triangleq \sum_{i=1}^m \sum_{j=1}^n \mathbb{I}\{X_{ij} \neq 0\}$). The parameter $k$ specifies the total budget.

Building on the LoRA formulation, we approximate the individual factors $A$ and $B$, aiming to find compressed counterparts $\tilde{A}$, $\tilde{B}$ such that:

$$\min_{\tilde{A}, \tilde{B}} \|BA - \tilde{B}\tilde{A}\|_F^2, \quad \text{s.t.} \ \|\tilde{A}\|_0 \leq k_A, \ \|\tilde{B}\|_0 \leq k_B, \quad k_A + k_B = k, \tag{2}$$

where $k_A$ and $k_B$ represent budgets for $\tilde{A}$ and $\tilde{B}$, respectively. This problem is challenging: counting the number of nonzero elements is non-convex, sparse element selection is combinatorial, and excessive sparsity may degrade accuracy. Achieving high compression without task performance loss thus requires careful subspace design and adaptive optimization.

## 2.2 METHOD: SUBSPACE-ORIENTED RANDOMIZED BASIS, SPARSE SELECTION, AND RECONSTRUCTION

To solve (2), we propose SOLAR. A key insight motivating our approach is that $\Delta W$ predominantly resides in the subspace spanned by $W$, particularly in LoRA-based fine-tuning, where constraining the rank $r \ll \min(m, n)$ forces $\Delta W$ to concentrate its variation along specific directions of $W$ Hu et al. (2021). This alignment (i.e., the overlap in the principal directions of $W$ and $\Delta W$) has been observed empirically and explained theoretically via neural tangent kernel (NTK) theory Jacot et al. (2018); Malladi et al. (2023); Seleznova et al. (2023). The left- and right-singular alignments are measured as $\|U_W^\top U_{\Delta W}\|_F^2$ and $\|V_W^\top V_{\Delta W}\|_F^2$, where $U$ and $V$ contain the left and right singular vectors from the SVD of each matrix Hu et al. (2021). Under this perspective, the model's response to updates is well-approximated by a first-order expansion: $f(\xi; W + \Delta W) \approx f(\xi; W) + \langle \nabla f(\xi; W), \Delta W \rangle$, where $f$ is the model, $\xi$ is input data, and $\nabla_W f(\xi; W)$ denotes the gradient of the foundation model's output. This implies that $\Delta W$ lies in a low-curvature (and hence low-dimensional) subspace defined by $W$'s parameter space (see Section 3.4 for empirical evidence). Thus, projecting $\Delta W$ into the subspace of $W$ enables an efficient and compact representation that can be sparsified with minimal information loss.

Building on these insights, we design a three-stage compression framework (Figure 1). First, we construct a randomized basis set aligned with the foundation model (Section 2.2.1). Next, we select a sparse set of bases to approximate the projected update (Section 2.2.2). We then reconstruct the update using a budget-aware combination of selected components (Section 2.2.3).

### 2.2.1 STEP 1: SUBSPACE-ORIENTED RANDOMIZED BASIS SET

We construct a basis set from the foundation model's parameter space via SVD of the model weight, $W = U\Sigma V^T$, where $U \in \mathbb{R}^{m \times m}$ and $V \in \mathbb{R}^{n \times n}$ are orthonormal, and $\Sigma \in \mathbb{R}^{m \times n}$ is diagonal. This

decomposition enables a basis naturally aligned with the directions of task-specific updates $\Delta W$. Unlike methods such as NOLA Koohpayegani et al. (2024) relying on unstructured random bases, our foundation-aligned directions allow a more compact representation of $\Delta W$.

To enrich the expressive power of this subspace, we construct randomized basis matrices by perturbing slices of the singular vectors:

$$\mathcal{M}_A = \left\{ M_A^{(i)} = V[:, \mathcal{I}_i] + \epsilon_i \right\}_{i=1}^{N_A}, \quad \mathcal{M}_B = \left\{ M_B^{(j)} = U[:, \mathcal{J}_j] + \epsilon_j \right\}_{j=1}^{N_B}, \quad (3)$$

where $\mathcal{I}_i$ and $\mathcal{J}_j$ are randomly sampled index sets, $N_A, N_B$ are the number of basis candidates for $A$ and $B$, respectively, and $\epsilon_i, \epsilon_j$ are random matrices with each entry drawn i.i.d. from $\mathcal{N}(0, 1)$. These basis sets form a flexible pool of candidates for approximation.

### 2.2.2 STEP 2: SPARSE SELECTION OF BASES

To enable more compact approximations, the LoRA update $\Delta W = BA$ is first projected into the subspace of $W$. Given the singular value decomposition $W = U\Sigma V^T$, this projection is defined as $\Delta W_{\text{Proj}} = U^T \Delta W V = (U^T B)(AV) = B_{\text{Proj}} A_{\text{Proj}}$, where $A_{\text{Proj}} = AV$ and $B_{\text{Proj}} = U^T B$ represent the update components expressed in the basis of $W$. This transformation retains all information when $W$ is full-rank, and is particularly effective when $\Delta W$ is already aligned with the foundation subspace, a property commonly observed in LoRA-based fine-tuning. Under this projection, the update becomes $\Delta W = U\Delta W_{\text{Proj}} V^T$. This approach leverages the inherent alignment between $W$ and $\Delta W$, enabling more efficient approximations with fewer basis elements than methods such as NOLA, which rely on unstructured random projections. Specifically, we approximate the projected LoRA factors $AV$ and $U^T B$ using sparse linear combinations of the basis matrices:

$$\min_{\alpha} \left\| AV - \sum_{i=1}^{N_A} \alpha_i M_A^{(i)} \right\|_F^2, \text{s.t. } \|\alpha\|_0 \leq k_A, \quad \min_{\beta} \left\| U^T B - \sum_{j=1}^{N_B} \beta_j M_B^{(j)} \right\|_F^2, \text{s.t. } \|\beta\|_0 \leq k_B. \quad (4)$$

A two-step strategy is employed to solve these NP-hard problems efficiently. The first step computes the unconstrained least squares solution to obtain coefficients $\alpha^*$ and $\beta^*$. The second step applies hard thresholding to retain only the top$_k$ entries by magnitude based on the budgets $k_A$ and $k_B$.

### 2.2.3 STEP 3: BUDGET-AWARE RECONSTRUCTION

The approximated model update is then reconstructed using the selected top$_k$ bases, resulting in $\tilde{A}$ and $\tilde{B}$ for $A$ and $B$, respectively:

$$A \approx \left( \sum_{i \in S_A} \alpha_i^* M_A^{(i)} \right) V^T, \qquad B \approx U \left( \sum_{j \in S_B} \beta_j^* M_B^{(j)} \right), \quad (5)$$

where $S_A$ and $S_B$ are the selected top$_k$ index sets. Because the update reconstruction is performed within the subspace defined by $W$, this step ensures strong alignment with task-relevant directions. The reconstruction balances accuracy and compression, with the sparsity budgets $k_A$ and $k_B$ controlling the number of active basis.

**Adaptive Compression.** SOLAR enables flexible allocation of sparsity budgets $k_A$ and $k_B$, adapting to system constraints such as memory, storage, or bandwidth. This allows deployment on resource-constrained devices, with adapter size dynamically adjustable post-training. For instance, a server can send a compact adapter to low-memory clients and a richer version to more capable devices.

### 2.3 THEORETICAL ANALYSIS OF RECONSTRUCTION ERROR

We assume that (A1) the model is initialized with spectral initialization; (A2) the optimal update is low-rank; (A3) the change in the model's weights from fine-tuning is well-behaved according to the generation process in Zhang et al. (2025a); and (A4) the singular values of the projected update matrix exhibit Fast Spectrum Decay. These assumptions are well-established and frequently utilized in the literature for convergence analyses, as in previous works, such as Zhang et al. (2025a); Martinsson & Tropp (2020).

**Theorem 1 [SOLAR Reconstruction Error Bound]** Let $\Delta W^*$ be the optimal low-rank adapter, $\Delta W$ be the adapter learned via fine-tuning, and $\Delta \tilde{W}$ be the adapter reconstructed by SOLAR. Under assumptions (A1)–(A4), the expected total error is bounded by $\mathbb{E}\left[\|\Delta \tilde{W} - \Delta W^*\|_F\right] \leq C_1 + C_2$, where $C_1$ captures the fine-tuning error (depending on learning rate, training steps, and spectrum of $\Delta W^*$; see Appendix A), and $C_2 = \sqrt{1 + \frac{r_A}{N_A - r_A - 1}} \left(\sum_{t > r_A} \sigma_t^2(\Delta W)\right)^{\frac{1}{2}} + \sqrt{1 + \frac{r_B}{N_B - r_B - 1}} \left(\sum_{t > r_B} \sigma_t^2(\Delta W)\right)^{\frac{1}{2}} + \left(\sum_{t > k} \sigma_t^2(\Delta W)\right)^{\frac{1}{2}}$, where $\sigma_t(\Delta W)$ is the $t$-th singular value of the fine-tuned update $\Delta W$, and $r_A, r_B$ denote the effective ranks after moving to the random basis space. The SOLAR reconstruction error has two parts: the fine-tuning error ($C_1$) and the compression error ($C_2$). The compression error decreases with larger basis pools ($N_A, N_B$) and higher sparsity budget ($k$). Details are in Appendix A.

## 3 EXPERIMENTS

We evaluate SOLAR through extensive experiments in three domains: 1) image classification with ViT-B/L in few-shot and full-data settings (Section 3.1); 2) instruction tuning on LLaMA-3 models using Alpaca and MMLU (Section 3.2); and 3) language generation with GPT-2 on E2E NLG (Section 3.3). Across all settings, SOLAR matches LoRA and NOLA in accuracy while reducing adapter size by up to 98%, offering a lightweight representation for model adaptation.

### 3.1 SOLAR ON VISION TRANSFORMERS

We conduct few-shot image classification experiments using ViT-B and ViT-L Dosovitskiy et al. (2020) foundation models, initialized with either supervised or self-supervised He et al. (2022).

**Experimental Setup.** We compare SOLAR against LoRA Hu et al. (2021) and NOLA Koohpayegani et al. (2024). Experiments are conducted on ViT-Base (ViT-B) and ViT-Large (ViT-L) architectures. Supervised ViT models pretrained on ImageNet-21k Deng et al. (2009) are obtained from Google's official releases via the Hugging Face repository Wolf et al. (2020); Research (2025), and MAE models pretrained on ImageNet-1K are sourced from the Timm library Wightman (2025). All experiments run on a single NVIDIA RTX 4090 GPU using PyTorch Paszke (2019) and HuggingFace libraries. In SOLAR, the compressed representation consists of (i) a random seed to regenerate the basis vectors, (ii) an encoded list of selected basis indices, and (iii) their coefficients. Reported trainable parameters include both projection coefficients and overhead (i.e., seed and index encoding). The MLP classifier head is dataset-specific and excluded from the parameter count unless noted.

**Evaluation Benchmarks.** We fine-tune on standard image classification datasets: CIFAR-10 Krizhevsky et al. (2009), CIFAR-100 Krizhevsky et al. (2009), Food-101 Bossard et al. (2014), Tiny-ImageNet Le & Yang (2015), ImageNet-1K Deng et al. (2009), Oxford Pets Parkhi et al. (2012), SUN397 Xiao et al. (2010), and CUB-200-2011 Welinder et al. (2010).

**Comparison Methods.** We compare SOLAR with several baselines: Full Fine-Tuning (Full-FT), LoRA Hu et al. (2021), and NOLA Koohpayegani et al. (2024). In Full-FT, all backbone parameters are updated. For LoRA, we apply low-rank adapters to the attention Query projection matrices, with a rank of 4 for ViT-B and either 1 or 4 for ViT-L. For NOLA, following Koohpayegani et al. (2024), adapters are inserted into MLP layers using 1000 random basis vectors for each of the $A$ and $B$ matrices. All models are trained with cross-entropy loss. For full-data settings, we train 5 epochs with batch size 128; for few-shot settings (10 samples per class), 25 epochs with batch size 16, emphasizing low-data efficiency relevant to real-world and distributed scenarios. To account for variance from limited data, we sample four training splits per dataset and report mean top-1 accuracy on the test split (or validation for ImageNet-1k). Experiments are repeated with different random seeds, and learning rates are tuned per dataset and model. Additional details are in the appendix.

**Results and Performance Analysis.** We evaluate SOLAR on various vision benchmarks using foundation models, with results in Table 1. In the tables, configurations are denoted as $\text{SOLAR}_{\text{method}(N \to k)}$, indicating that SOLAR is applied to a NOLA or LoRA model trained with rank $r$, using $N$ bases per matrix ($N = N_A = N_B$) and selecting the top-$k$ bases by significance, where $N$ and $k$ are given in thousands. SOLAR consistently achieves competitive top-1 accuracy in few-shot (10 samples per

Table 1: Top-1 classification accuracy (%) of ViT-B and ViT-L on benchmark datasets under two settings: (1) few-shot (10 samples/class, 25 epochs) and (2) full-data (5 epochs). Results report mean $\pm$ std over 5 runs. SOLAR is applied with configuration $_{\text{method}(N \to k)}$, where $N$ and $k$ are in thousands.

| Model | Method | # Param | CIFAR-10 | | CIFAR-100 | | Food-101 | | T-ImageNet | |
|---|---|---|---|---|---|---|---|---|---|---|
| | | | 10 | Full | 10 | Full | 10 | Full | 10 | Full |
| ViT-B | Full-FT | 86M | $91.1_{\pm.8}$ | $94.6_{\pm.5}$ | $78.2_{\pm.7}$ | $87.7_{\pm.3}$ | $65.8_{\pm.9}$ | $85.2_{\pm.4}$ | $78.1_{\pm1.0}$ | $85.4_{\pm.6}$ |
| | LoRA ($r$=4) | 74K | $\mathbf{92.3}_{\pm.6}$ | $\mathbf{98.3}_{\pm.2}$ | $\mathbf{81.8}_{\pm.8}$ | $\mathbf{90.3}_{\pm.4}$ | $\mathbf{72.4}_{\pm.7}$ | $\mathbf{87.6}_{\pm.3}$ | $77.9_{\pm.9}$ | $\mathbf{88.8}_{\pm.4}$ |
| | NOLA | 48K | $92.2_{\pm.6}$ | $94.7_{\pm.5}$ | $81.3_{\pm.8}$ | $86.6_{\pm.4}$ | $\underline{72.6}_{\pm.5}$ | $85.9_{\pm.2}$ | $\mathbf{78.4}_{\pm.7}$ | $82.8_{\pm.5}$ |
| | SOLAR$_{r=4(4\to1.6)}$ | $\underline{41K}$ | $\mathbf{92.3}_{\pm.7}$ | $\mathbf{98.3}_{\pm.4}$ | $\underline{81.5}_{\pm.7}$ | $\underline{89.8}_{\pm.2}$ | $71.8_{\pm.6}$ | $\underline{87.0}_{\pm.5}$ | $77.9_{\pm.8}$ | $\underline{87.9}_{\pm.4}$ |
| | SOLAR$_{\text{NOLA}(4\to1.2)}$ | $\mathbf{32K}$ | $92.1_{\pm.7}$ | $94.5_{\pm.3}$ | $81.1_{\pm.6}$ | $85.4_{\pm.3}$ | $72.5_{\pm.6}$ | $85.4_{\pm.3}$ | $\underline{78.3}_{\pm.8}$ | $82.3_{\pm.5}$ |
| ViT-L | Full-FT | 303M | $90.2_{\pm.9}$ | $94.1_{\pm.6}$ | $86.2_{\pm.7}$ | $87.7_{\pm.5}$ | $73.9_{\pm.8}$ | $85.5_{\pm.4}$ | $80.8_{\pm1.1}$ | $89.2_{\pm.6}$ |
| | LoRA ($r$=4) | 197K | $\mathbf{97.1}_{\pm.5}$ | $\mathbf{98.7}_{\pm.1}$ | $\mathbf{88.1}_{\pm.7}$ | $\underline{92.4}_{\pm.3}$ | $81.8_{\pm.7}$ | $\mathbf{89.8}_{\pm.2}$ | $\mathbf{84.4}_{\pm.8}$ | $\mathbf{91.8}_{\pm.5}$ |
| | LoRA ($r$=2) | 98K | $\underline{96.6}_{\pm.4}$ | $\mathbf{98.7}_{\pm.1}$ | $\mathbf{88.0}_{\pm.6}$ | $\mathbf{92.9}_{\pm.3}$ | $\underline{82.1}_{\pm.7}$ | $\mathbf{90.0}_{\pm.2}$ | $83.8_{\pm.7}$ | $\underline{90.4}_{\pm.3}$ |
| | NOLA | 96K | $96.0_{\pm.8}$ | $97.4_{\pm.6}$ | $87.8_{\pm1.0}$ | $89.3_{\pm.5}$ | $\mathbf{82.5}_{\pm.8}$ | $86.7_{\pm.4}$ | $\mathbf{84.3}_{\pm.9}$ | $86.7_{\pm.6}$ |
| | SOLAR$_{r=4(4\to1.6)}$ | 82K | $\mathbf{97.0}_{\pm.5}$ | $\underline{98.5}_{\pm.3}$ | $\underline{87.9}_{\pm.8}$ | $91.4_{\pm.4}$ | $76.8_{\pm.7}$ | $\underline{87.1}_{\pm.4}$ | $78.7_{\pm.7}$ | $88.6_{\pm.5}$ |
| | SOLAR$_{r=2(1\to0.3)}$ | $\mathbf{50K}$ | $96.1_{\pm.8}$ | $98.2_{\pm.4}$ | $87.4_{\pm.9}$ | $90.0_{\pm.5}$ | $77.0_{\pm.8}$ | $86.8_{\pm.6}$ | $76.4_{\pm.9}$ | $87.6_{\pm.6}$ |
| | SOLAR$_{\text{NOLA}(4\to1.2)}$ | $\underline{64K}$ | $95.8_{\pm.9}$ | $97.0_{\pm.4}$ | $87.7_{\pm.8}$ | $89.3_{\pm.4}$ | $\underline{82.1}_{\pm.7}$ | $86.6_{\pm.3}$ | $\underline{84.1}_{\pm.8}$ | $86.4_{\pm.6}$ |

Table 2: Additional evaluation on vision datasets using ViT-B. The table shows bit-level representation footprint (32-bit baseline) and top-1 accuracy. All models are trained for 10 epochs.

| Method | Byte Footprint | Oxford Pets | SUN397 | CUB-200 | ImageNet-1K |
|---|---|---|---|---|---|
| LoRA ($r$=1) | 74KB | $\mathbf{93.0}_{\pm-.3}$ | $\mathbf{74.3}_{\pm-.2}$ | $\mathbf{84.7}_{\pm-.2}$ | $\mathbf{81.5}_{\pm-.4}$ |
| NOLA | 48KB | $90.4_{\pm-.5}$ | $61.7_{\pm-.4}$ | $79.4_{\pm-.4}$ | $77.4_{\pm-.3}$ |
| SOLAR$_{r=1(2\to0.2)}$ | $\mathbf{8KB}$ (89% $\downarrow$) | $\underline{92.6}_{\pm-.4}$ | $\underline{73.9}_{\pm-.2}$ | $\underline{84.2}_{\pm-.3}$ | $\underline{81.3}_{\pm-.2}$ |

Table 3: Effect of quantization on SOLAR$_{r=4(4\to1.6)}$ performance. ViT-L-MAE fine-tuned on CIFAR-10.

| Method | Quant. | Accuracy | Byte Footprint |
|---|---|---|---|
| SOLAR | 32-bit | $86.7_{\pm-.3}$ | 319KB |
| | 16-bit | $86.5_{\pm-.3}$ | 166KB |
| | 8-bit | $85.9_{\pm-.4}$ | 89KB |
| | 4-bit | $84.8_{\pm-.6}$ | 50KB |

Table 4: Effect of rank and adapter placement in SOLAR$_{r=4(4\to1)}$. Accuracy (%) on CIFAR-100 using ViT-B.

| Rank | Q | K | V | QV | QKV |
|---|---|---|---|---|---|
| 1 | 87.0 | 85.5 | 86.6 | 88.3 | 90.1 |
| 2 | 87.5 | 85.7 | 87.4 | 88.6 | 90.5 |
| 4 | 87.8 | 86.1 | 87.5 | 89.0 | 90.6 |
| 8 | 88.1 | 86.0 | 87.4 | 89.1 | 90.7 |
| 16 | 87.9 | 86.0 | 87.1 | 89.0 | 90.6 |

class) and full-data settings while requiring far fewer trainable parameters than LoRA and NOLA. On ViT-B and ViT-L, SOLAR matches LoRA's performance using up to 74% fewer parameters. For instance, applied to a LoRA ($r = 2$), bases $N_A = N_B = 4000$, and top$_k = 1600$, SOLAR reduces fine-tuned parameters from 98K to 25K while maintaining comparable accuracy.

Beyond parameter reduction, SOLAR improves storage efficiency. Table 2 reports mean and standard deviation over 5 runs on four additional datasets using ViT-B, quantifying the bit-level footprint assuming 32-bit precision during training. We apply 8-bit quantization to SOLAR after top$_k$ parameter selection. While LoRA ($r = 1$) requires 74KB of adapter parameters, SOLAR reduces this to 8KB (89% reduction). These extreme compressions incur only minor accuracy drops, showing SOLAR enables fine-grained control of model size to meet strict constraints and offers a flexible tradeoff between footprint and performance.

In addition to reducing parameter and storage footprints, SOLAR remains highly robust under quantization. As shown in Table 3, reducing coefficient precision from 32-bit to 4-bit incurs less than a 2% accuracy drop on ViT-L-MAE (CIFAR-10, 10-shot). We further evaluate the effect of adapter rank and placement (Table 4), observing that performance improves with rank up to 8 (with higher ranks requiring more time to converge), and that the Query (Q) projection yields the highest gains.

Table 5: Model representation efficiency for LLaMA models. SOLAR compresses LoRA adapter updates across various model sizes. For the 13B model, all methods use 4-bit quantization, making the LoRA baseline equivalent to QLoRA.

| Model | LLaMA-3.2 1B | | | LLaMA-2 13B (4-bit) | | |
|---|---|---|---|---|---|---|
| Method | LoRA $r$=8 | NOLA 1000 bases | SOLAR $r = 8(4 \rightarrow 1.2)$ | LoRA $r$=1 | NOLA 1000 bases | SOLAR $r = 1(1 \rightarrow 0.3)$ |
| # Params | 852K | **64K** | 81K (90% ↓) | 819K | 140K | **51K** (94% ↓) |
| Val Loss | **1.51** | 1.87 | **1.52** | 1.05 | 1.29 | **1.05** |
| MMLU Acc | **30.1** | 25.9 | 28.3 | **54.5** | 51.8 | **54.5** |

Table 6: Performance and parameter efficiency on E2E NLG using GPT-2 Small and Medium. All methods use rank-4 adapters applied to the Query and Value projections.

| Method | GPT-2 Small | | GPT-2 Medium | |
|---|---|---|---|---|
| | MET | # Params | MET | # Params |
| Full-FT | 28.4 | 124M | 46.2 | 355M |
| LoRA ($r$=4) | **29.7** | 147K | **47.2** | 393K |
| NOLA | 29.1 | 48K | 46.8 | 350K |
| SOLAR ($r$=4, 1→0.3) | **29.7** | 15K (90% ↓) | 46.4 | 30K (92% ↓) |
| SOLAR ($r$=1, 0.1→0.1) | 26.1 | **4K** (97% ↓) | 44.8 | **9K** (98% ↓) |

## 3.2 SOLAR ON LLAMA

**Experimental Setup.** We apply SOLAR to LLaMA-3 models of size 1B–13B. All models are fine-tuned using adapters in the query and value projections across all transformer layers. For the 1B model, we use LoRA with rank 8; for the 31B model, we use LoRA with rank 1. To reduce GPU memory usage for large-scale models, we quantize the 13B model using 4-bit NF4 quantization through the `BitsAndBytes` library Dettmers et al. (2021); Dettmers (2025). Further implementation details and hardware configurations are provided in the Appendix.

**Evaluation Benchmarks.** All models are fine-tuned on the Stanford Alpaca Taori et al. (2023) dataset for instruction-following and evaluated on its validation loss. We also assess generalization to out-of-distribution tasks using the MMLU benchmark Hendrycks et al. (2020).

**Comparison Methods.** We compare SOLAR with PEFT baselines, including LoRA Hu et al. (2021) and NOLA Koohpayegani et al. (2024). LoRA uses rank $r = 8$ for LLaMA-3 1B and $r = 1$ for the 13B model. NOLA follows its original configuration, with 1000 random basis vectors per matrix Koohpayegani et al. (2024). For the 13B model, we apply 4-bit quantization to all methods (LoRA, NOLA, and SOLAR). The reported trainable parameters include learned coefficients and overhead for basis indexing. All experiments use gradient checkpointing, and learning rates are tuned separately per model and method to ensure a fair comparison.

**Results and Performance Analysis.** Table 5 reports results across model sizes. SOLAR matches LoRA in Alpaca validation loss and MMLU Hendrycks et al. (2020) accuracy while reducing trainable adapter parameters by up to 94%. For example, on LLaMA-3.2 13B, SOLAR cuts the adapter size from 819K to 51K without accuracy loss.

## 3.3 SOLAR ON GPT-2

**Experimental Setup.** We evaluate our method on GPT-2 Radford et al. (2019) base and medium models fine-tuned on the E2E NLG dataset Novikova et al. (2017) using LoRA. The models are trained for 5 epochs using a batch size of 8 and a learning rate of 0.1. LoRA is applied to the self-attention Query and Value projection, with a rank of $r = 4$. After training, we apply SOLAR to compress the LoRA adapter updates.

**Evaluation Benchmarks.** We use the E2E NLG dataset to evaluate generative quality. Generated outputs are assessed using METEOR Banerjee & Lavie (2005) metric. We report LoRA, NOLA, and SOLAR performance.

**Results and Performance Analysis.** Table 6 summarizes results on the E2E NLG dataset using GPT-2 Small and Medium models. SOLAR achieves competitive METEOR scores compared to LoRA and NOLA, while substantially reducing adapter size. On GPT-2 Medium, SOLAR reduces adapter representation size from 393K (LoRA) to 30K parameters with minimal performance loss. Applied to rank-1 LoRA, it achieves a 98% reduction, demonstrating strong compression capability.

### 3.4 DISCUSSION AND ANALYSIS ON SOLAR PERFORMANCE AND EFFICIENCY

**Subspace Analysis.** We analyze the subspace similarity between the foundation model's weights $W$ and the LoRA update $\Delta W$ with rank $r = 4$ (see Figure 2). Let $W = U_W \Sigma_W V_W^\top$ and $\Delta W = U_{\Delta W} \Sigma_{\Delta W} V_{\Delta W}^\top$ denote their SVDs. To quantify subspace alignment, we define the similarity function as $\phi(W, \Delta W, i, j) = \psi(U_W^{(i)}, U_{\Delta W}^{(j)}) = \|U_W^{(i)\top} U_{\Delta W}^{(j)}\|_F^2$, where $U_W^{(i)}$ and $U_{\Delta W}^{(j)}$ are the matrices formed by taking the $i$ and $j$ left singular vectors of $W$ and $\Delta W$, respectively. This normalized Frobenius inner product measures how much of the $j$-dimensional subspace of $\Delta W$ lies within the $i$-dimensional subspace of $W$, reaching its maximum when perfectly aligned. Figure 2 shows

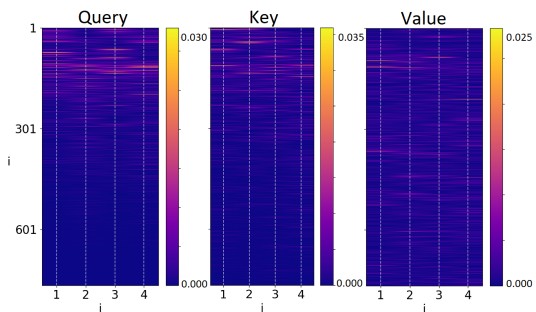

Figure 2: Subspace similarity between the $W$ and $\Delta W$ matrices (Q, K, V) from the first layer of the ViT-B model using LoRA with rank $r = 4$.

that the fine-tuned model emphasizes directions already present in the foundation model, supporting prior observations that LoRA updates lie in low-dimensional, structured subspaces Hu et al. (2021); Farhadzadeh et al. (2025); Zhang et al. (2025b). This suggests leveraging existing directions is more effective than relying purely on random ones: LoRA implicitly aligns with them, and SOLAR exploits this alignment in its basis pool, explaining its performance advantage over NOLA.

**Effect of Basis Pool Size and Communication Budget on Performance.** To evaluate SOLAR's trade-off between representation size and performance, we analyze the effect of varying the basis pool size and the number of selected $\text{top}_k$ components on representation accuracy. Experiments are conducted on a ViT-Base model fine-tuned using LoRA with rank 4, followed by SO-LAR compression. Each LoRA matrix $A$ and $B$ requires $4 \times 768 = 3072$ parameters. We observe that increasing $k$ improves SOLAR's expressiveness and accuracy. Moreover, a larger basis pool enhances performance by increasing the likelihood of capturing directions aligned with the fine-tuned model subspace. As shown

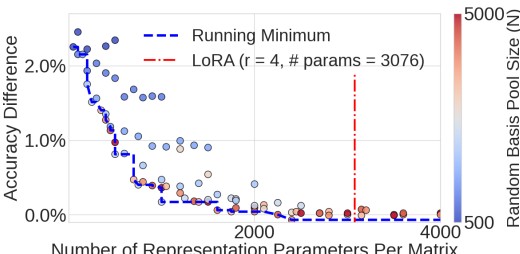

Figure 3: Representation Performance vs. Representation Cost: On ViT-B ($r = 4$), SOLAR demonstrates a trade-off between parameter count and performance, achieving strong performance with far fewer parameters than LoRA.

in Figure 3, even with fixed $k$, larger pools yield higher accuracy by enabling more precise reconstruction of target directions. SOLAR thus achieves performance comparable to LoRA with significantly fewer parameters. This trade-off confirms Theorem 1: increasing the basis pool $N$ or sparsity $k$ reduces the compression error $C_2$.

**SOLAR Overhead and Runtime Efficiency.** As a post-training method, SOLAR introduces negligible runtime overhead and does not interfere with fine-tuning. For instance, fine-tuning LLaMA-3.2 1B with LoRA on Tiny-ImageNet took 2081 seconds, while SOLAR, including random basis generation, convex least-squares solving, and $\text{top}_k$ selection, took only 15 seconds (under 0.72% of training time). These operations are computationally lightweight, as shown in Table 7, confirming SOLAR's practical efficiency.

Table 7: Runtime Overhead: LoRA (10 epochs) vs. SOLAR post-training on ViT-B across vision datasets. Times in seconds.

| Dataset | LoRA | SOLAR | Overhead (%) |
|---|---|---|---|
| CIFAR-10 | 1176 | 14 | 1.19 |
| CIFAR-100 | 1165 | 14 | 1.20 |
| Food-101 | 3480 | 67 | 1.92 |
| Tiny-ImageNet | 2081 | 15 | 0.72 |
| ImageNet-1K | 56634 | 155 | 0.27 |

**Limitations and Future Work.** As a post-hoc method, SOLAR's performance is limited by the base adapter, and its hyperparameters ($N$ and $k$) may need per-task tuning to optimize the compression-accuracy trade-off. While it shows strong results on vision and language tasks, its effectiveness on other modalities (audio, time series, or multimodal data) remains untested. Future work will extend SOLAR to these areas and evaluate its performance in other environments.

## 4 BACKGROUND AND RELATED WORKS

**Transformers in NLP and Vision.** Transformers Vaswani et al. (2017), are now the standard in NLP for modeling long-range dependencies via self-attention Raiaan et al. (2024). Models such as LLaMA Touvron et al. (2023), BERT Devlin et al. (2019), and GPT Radford et al. (2018) build on this structure to achieve strong results across diverse benchmarks. In vision, ViT Dosovitskiy et al. (2020) treats image patches as tokens, making Transformers a unifying backbone across modalities.

**Parameter-Efficient Fine-Tuning (PEFT).** As transformers scale, task-specific fine-tuning becomes computationally intensive. PEFT methods mitigate this by updating only a subset of parameters. LoRA Hu et al. (2021) introduces trainable low-rank matrices per layer, typically modifying <1% of weights, while NOLA Koohpayegani et al. (2024) re-parameterizes these as linear combinations of random bases, decoupling parameters from rank and architecture. Yet PEFT gains often fall short in deployment, especially on edge, mobile, and federated settings with communication and storage bottlenecks. Adapting GPT-2 (117M) on-device may still require gigabytes of transfer and petaflop-scale computation per round Wang et al. (2025), with updates taking seconds to transmit and hours to process on low-power hardware (e.g., Jetson TX2).

**Challenges of PEFT.** As models grow, adapter overhead scales rapidly. Even modest adapters (e.g., 7M parameters for a 7B model at rank 16) accumulate significant costs across users, tasks, or training rounds Xu et al. (2023b). A 1% adapter for LLaMA-2 70B adds 700M parameters; for GPT-3 (350B), 3.5B—tens of gigabytes in FP32. Such costs are infeasible in personalized or federated settings, where hundreds of adapters may be exchanged or stored per user Zhang et al. (2024). While PEFT leverages the low intrinsic dimensionality of task adaptation Hu et al. (2021), deployment remains inefficient. It has been shown that BERT fine-tuning on MRPC Dolan & Brockett (2005) requires only 1,861 degrees of freedom out of 110M, highlighting redundancy in full-rank updates Aghajanyan et al. (2020). Yet even small adapters impose substantial overhead on massive models Xu et al. (2023a); Lialin et al. (2023). Hence, the true bottleneck is adapter size, not fine-tuning efficiency Jie et al. (2023), motivating flexible post-training compression to reduce footprint without altering training.

**PEFT Compression Techniques.** To mitigate PEFT costs, pruning Han et al. (2024); Ilhan et al. (2024) and quantization Chen et al. (2024); Hubara et al. (2021) have been explored. These reduce model size but require careful tuning or retraining, are less effective under severe bandwidth limits, and are mainly optimized for full-model compression, limiting applicability to adapters. Adapter updates are highly redundant and lie in low-dimensional subspaces Hu et al. (2021); Yadav et al. (2023); Wu et al. (2024), motivating post-training compression. Methods like ComPEFT Yadav et al. (2023), BitDelta Liu et al. (2024), Delta-CoMe Ping et al. (2024), and DeltaZip Yao et al. (2025) compress adapter weights after fine-tuning but rely on heuristics, task-specific tuning, or training integration, reducing flexibility. Other approaches alter fine-tuning itself: VeRA Kopiczko et al. (2023) employs a shared random basis, SVFT Lingam et al. (2024) learns sparse coefficients for an SVD-based basis, and EigenLoRAx Kaushik et al. (2025) builds a PCA basis from many pre-trained adapters. In contrast, SOLAR is a post-hoc, training-free utility that compresses any adapter, providing a complementary plug-and-play solution.

## 5 CONCLUSION

Adapter-based fine-tuning methods such as LoRA significantly reduce the cost of adapting large models. However, in distributed and on-device settings, communication and storage overheads remain a major bottleneck. To address this, we introduce SOLAR, a lightweight post-training compression method that reparameterizes adapter updates as sparse combinations of structured basis vectors aligned with the foundation model's latent subspace. SOLAR substantially reduces adapter size and transmission cost without altering the training process or model architecture.

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

APPENDIX

## A PROOF OF THEOREM 1

Let $\Delta W^* \in \mathbb{R}^{m \times n}$ denote the optimal adapter for the downstream task, $\Delta W$ the adapter obtained by LoRA fine-tuning, and $\Delta \widetilde{W}$ the SOLAR reconstruction. Let $\Delta W_{\mathrm{proj}}$ denote the projection of $\Delta W$ onto the SOLAR bases (i.e., bases that are constructed from the SVD of the foundation model's weights, combined with randomized perturbations).

Our proof relies on the following standard assumptions from the literature on parameter-efficient fine-tuning and randomized numerical linear algebra:

(A1) *Spectral Initialization:* The LoRA adapter matrices $A$ and $B$ are initialized using the spectral initialization strategy from Zhang et al. (2025a).

(A2) *Low-Rank Update:* The optimal task-specific update $\Delta W^*$ is approximately low-rank, with rank $r^* < \min\{m, n\}$ Zhang et al. (2025a).

(A3) *Well-Behaved Data:* The training data follows the generation process outlined in Zhang et al. (2025a), where input features are drawn from an isotropic sub-Gaussian or Gaussian distribution.

(A4) *Fast Spectrum Decay:* The projected update matrix $\Delta W_{\mathrm{proj}}$ exhibits spectral decay, meaning its tail singular values are small (Martinsson & Tropp, 2020).

First, we decompose the total error using the triangle inequality. The total error, $\|\Delta \widetilde{W} - \Delta W^*\|_F$, is the distance between the SOLAR-reconstructed adapter and the optimal adapter. This is bounded by the sum of the Training Error and the Compression Error:

$$\|\Delta \widetilde{W} - \Delta W^*\|_F \leq \underbrace{\|\Delta \widetilde{W} - \Delta W\|_F}_{\text{Compression Error}} + \underbrace{\|\Delta W - \Delta W^*\|_F}_{\text{Training Error}} \tag{6}$$

Here, the first term, $\|\Delta \widetilde{W} - \Delta W\|_F$, is the compression error introduced by SOLAR's approximation. The second term, $\|\Delta W - \Delta W^*\|_F$, is the training error from the underlying LoRA fine-tuning process itself. We will bound each term separately.

The analysis of the training error for LoRA adapters is non-trivial and has been extensively studied. We directly leverage the results from Zhang et al. (2025a), showing that under Assumptions (A1)-(A3), LoRA trained with gradient descent converges to the optimal low-rank adapter $\Delta W^*$. Their analysis provides the following bound on the training error after $t$ steps:

$$\|\Delta W - \Delta W^*\|_F \leq \sqrt{2r^*} \left(1 - \frac{\eta \lambda_{r^*}}{64\kappa}\right)^t \lambda_{r^*}, \tag{7}$$

where $r^*$ is the rank of the optimal update $\Delta W^*$, $\kappa$ is its condition number, $\lambda_{r^*}$ is its $r^*$-th singular value, and $\eta$ is the learning rate. This bound, derived under the specified spectral initialization and data concentration assumptions, demonstrates that the fine-tuned adapter $\Delta W$ gets exponentially closer to the optimal adapter $\Delta W^*$ as training progresses.

SOLAR reconstructs the adapter as a sparse coefficientization over these perturbed bases:

$$\Delta \widetilde{W} = \sum_{i=1}^{N_B} \sum_{j=1}^{N_A} \beta_i \alpha_j \, M_B^{(i)} M_A^{(j)}. \tag{8}$$

Following the randomized rangefinder formulation Halko et al. (2011); Martinsson & Tropp (2020), we construct the sketch matrices for both the column and row spaces of the LoRA-style adapter update $\Delta W$ as

$$Y_A = \Delta W \, \Omega_A \, \in \, \mathbb{R}^{m \times N_A}, \qquad Y_B = \Delta W^\top \Omega_B \, \in \, \mathbb{R}^{n \times N_B}. \tag{9}$$

Each column of $Y_A$ represents the action of $\Delta W$ on a random probe vector drawn from the right-basis pool $\Omega_A$, effectively sampling the column space of $\Delta W$. Similarly, each column of $Y_B$ captures

random projections of the row space of $\Delta W$. These sketches compactly encode the dominant directions of $\Delta W$ without explicitly computing its singular value decomposition.

The Gaussian perturbations in $M_A^{(i)} = V_{:,\mathcal{I}_i} + \epsilon_i$ and $M_B^{(j)} = U_{:,\mathcal{J}_j} + \epsilon_j$ play an important theoretical and practical role. First, they ensure that the composite sketching matrices $\Omega_A$ and $\Omega_B$ satisfy the sub-Gaussian concentration and Johnson–Lindenstrauss properties required for the probabilistic error bounds in randomized numerical linear algebra Halko et al. (2011). Second, adding small isotropic noise expands the effective span of the sampled singular directions, preventing over-alignment with any single dominant mode and improving numerical stability when the singular spectrum of $\Delta W$ decays slowly. Finally, this perturbation acts as a regularizer that mitigates sampling bias inherited from the foundation model's specific singular subspace, ensuring broader coverage of the subspace where fine-tuned updates lie.

We then compute orthonormal bases for the column spans of these sketches:

$$Q_A = \text{orth}(Y_A) \in \mathbb{R}^{m \times q_A}, \qquad Q_B = \text{orth}(Y_B) \in \mathbb{R}^{n \times q_B}, \tag{10}$$

where

$$r_A = \text{rank}(Q_A) \leq \min(m, N_A), \qquad r_B = \text{rank}(Q_B) \leq \min(n, N_B).$$

By construction, $\text{range}(Q_A) = \text{range}(Y_A)$ and $\text{range}(Q_B) = \text{range}(Y_B)$. In the terminology of randomized numerical linear algebra, this process corresponds to the *rangefinder step*, which identifies low-dimensional subspaces that approximate the dominant column and row spaces of $\Delta W$.

Finally, we define the two-sided (bi-rangefinder) projection as

$$\mathcal{P}_{N_A, N_B}(\Delta W) := Q_A Q_A^\top \Delta W \, Q_B Q_B^\top. \tag{11}$$

This projection provides a low-rank approximation to $\Delta W$ using orthonormal subspaces inferred from randomized sketches. Geometrically, $\mathcal{P}_{N_A, N_B}(\Delta W)$ captures the principal subspace of $\Delta W$ identified by $\Omega_A$ and $\Omega_B$, offering an efficient surrogate for the optimal SVD-based projection $U_1 U_1^\top \Delta W V_1 V_1^\top$ while retaining probabilistic error guarantees Halko et al. (2011); Martinsson & Tropp (2020).

We bound the bi-projection error by splitting it into two one-sided parts using projector non-expansiveness ($\|Q_A Q_A^\top X\|_F \leq \|X\|_F$):

$$\|\Delta W - Q_A Q_A^\top \Delta W Q_B Q_B^\top\|_F \leq \|\Delta W - Q_A Q_A^\top \Delta W\|_F + \|Q_A Q_A^\top(\Delta W - \Delta W Q_B Q_B^\top)\|_F$$
$$\leq \|\Delta W - Q_A Q_A^\top \Delta W\|_F + \|\Delta W - \Delta W Q_B Q_B^\top\|_F. \tag{12}$$

Each addend is a standard one-sided rangefinder error. By Theorem 10.5 of Halko et al. (2011) (Frobenius form) with oversampling $N_A > r_A + 1$ and $N_B > r_B + 1$,

$$\mathbb{E} \|\Delta W - Q_A Q_A^\top \Delta W\|_F \leq \left(1 + \frac{r_A}{N_A - r_A - 1}\right)^{\frac{1}{2}} \left(\sum_{t > r_A} \sigma_t(\Delta W)^2\right)^{\frac{1}{2}}, \tag{13}$$

$$\mathbb{E} \|\Delta W - \Delta W Q_B Q_B^\top\|_F \leq \left(1 + \frac{r_B}{N_B - r_B - 1}\right)^{\frac{1}{2}} \left(\sum_{t > r_B} \sigma_t(\Delta W)^2\right)^{\frac{1}{2}}. \tag{14}$$

Combining equation 12–equation 14 yields the expected two-sided projection error bound:

$$\mathbb{E} \|\Delta W - \mathcal{P}_{N_A, N_B}(\Delta W)\|_F \leq \left(1 + \frac{r_A}{N_A - r_A - 1}\right)^{\frac{1}{2}} \left(\sum_{t > r_A} \sigma_t^2\right)^{\frac{1}{2}} + \left(1 + \frac{r_B}{N_B - r_B - 1}\right)^{\frac{1}{2}} \left(\sum_{t > r_B} \sigma_t^2\right)^{\frac{1}{2}}. \tag{15}$$

(When desired, power iterations can be incorporated on either side to sharpen the spectral decay and constants Halko et al. (2011); Martinsson & Tropp (2020).)

After projection, SOLAR enforces sparsity by retaining only the top-$k$ basis pairs in equation 8. Let the singular values of $\mathcal{P}_{N_A, N_B}(\Delta W)$ be $\{\tilde{\sigma}_t\}$, we have:

$$\|\Delta \widetilde{W} - \mathcal{P}_{N_A, N_B}(\Delta W)\|_F \leq \left(\sum_{t > k} \tilde{\sigma}_t^2\right)^{\frac{1}{2}}. \tag{16}$$

Moreover, orthogonal projections are contractions in Frobenius norm and cannot increase tail energy, hence

$$\sum_{t>k} \tilde{\sigma}_t^2 \;\le\; \sum_{t>k} \sigma_t(\Delta W)^2. \tag{17}$$

Adding and subtracting $\mathcal{P}_{N_A,N_B}(\Delta W)$ and using equation 15–equation 17, we obtain

$$\mathbb{E}\,\|\Delta \widetilde{W} - \Delta W\|_F \le \mathbb{E}\,\|\Delta W - \mathcal{P}_{N_A,N_B}(\Delta W)\|_F \;+\; \mathbb{E}\,\|\Delta \widetilde{W} - \mathcal{P}_{N_A,N_B}(\Delta W)\|_F$$

$$\le \left(1 + \frac{r_A}{N_A - r_A - 1}\right)^{\frac{1}{2}} \left(\sum_{t>r_A} \sigma_t^2\right)^{\frac{1}{2}} + \left(1 + \frac{r_B}{N_B - r_B - 1}\right)^{\frac{1}{2}} \left(\sum_{t>r_B} \sigma_t^2\right)^{\frac{1}{2}} \tag{18}$$

$$+ \left(\sum_{t>k} \sigma_t^2\right)^{\frac{1}{2}}. \tag{19}$$

Combining the decomposition with equation 19 and the LoRA training bound equation 7, we conclude

$$\mathbb{E}\,\|\Delta \widetilde{W} - \Delta W^*\|_F \;\le\; \underbrace{\left(1 + \frac{r_A}{N_A - r_A - 1}\right)^{\frac{1}{2}} \left(\sum_{t>r_A} \sigma_t^2\right)^{\frac{1}{2}} + \left(1 + \frac{r_B}{N_B - r_B - 1}\right)^{\frac{1}{2}} \left(\sum_{t>r_B} \sigma_t^2\right)^{\frac{1}{2}}}_{\text{projection error}}$$

$$+ \underbrace{\left(\sum_{t>k} \sigma_t^2\right)^{\frac{1}{2}}}_{\text{sparsification error}} + \underbrace{\sqrt{2r^*}\left(1 - \frac{\eta \lambda_{r^*}}{64\kappa}\right)^t \lambda_{r^*}}_{\text{training error}}. \tag{20}$$

Each term in equation 20 can be driven to zero under mild conditions: (i) the projection error vanishes as $N_A, N_B$ grow so that $r_A, r_B$ reach the true (or effective) rank of $\Delta W$ (then the corresponding spectral tails are zero); (ii) the sparsification error vanishes when $k$ exceeds the numerical rank of $\mathcal{P}_{N_A,N_B}(\Delta W)$; and (iii) the training error decays to zero as $t \to \infty$ under (A1)–(A3) by equation 7. Consequently, with sufficient sampling $(N_A, N_B)$, sparsity budget $(k)$, $\mathbb{E}\,\|\Delta \widetilde{W} - \Delta W^*\|_F \to 0$.

# B  IMPLEMENTATION DETAILS

All models are implemented using PyTorch Paszke (2019), with HuggingFace Transformers Wolf et al. (2020) for LLaMA and GPT-based models, and Timm Wightman (2025) for ViT-based vision backbones. Training and evaluation are performed on NVIDIA A100 and RTX 4090 GPUs. For all vision experiments, we use ViT-B and ViT-L as base encoders. For language models, we use GPT-2 and LLaMA-3 (1B, 3B, 8B). LoRA is applied to the query and value projections. SOLAR operates post-training by compressing the PEFT adapter matrices. All experiments are conducted under a fixed random seed for reproducibility. The implementation code for SOLAR, along with scripts used to reproduce the experiments, is included in the supplementary material and also available at `https://anonymous.4open.science/r/SOLAR-D3B2/`.

# C  DATASET DETAILS

We summarize dataset statistics in Table 8, including number of training samples and class counts.

We summarize dataset statistics used in the LLM experiments in Table 9, covering instruction tuning (Section 3.2) and language generation tasks (Section 3.3). The table includes the number of training samples, average sequence lengths, and the model-specific context in which each dataset is used in the experiments.

Table 8: Dataset statistics used in experiments. Each dataset includes the number of training samples and classes.

| Dataset | Training Samples | Number of Classes |
|---|---|---|
| CIFAR-10 | 50,000 | 10 |
| CIFAR-100 | 50,000 | 100 |
| Food-101 | 75,750 | 101 |
| Tiny-ImageNet | 100,000 | 200 |
| ImageNet-1K | 1,281,167 | 1,000 |

Table 9: Dataset statistics in LLM experiments.

| Dataset | Samples | Avg. Seq. Length | Context |
|---|---|---|---|
| Stanford Alpaca | 52,000 | ~256 tokens | LLaMA-3 instruction tuning |
| MMLU | 15,858 | ~200 tokens | LLaMA-3 Generalization evaluation |
| E2E NLG | 42,000 | ~35 tokens | GPT-2 generation fine-tuning |

# D    REPRESENTATION COST DETAILS: PARAMETERS AND STORAGE

To quantify SOLAR's compression benefit, we detail the number of adapter parameters and byte-level footprint across ViT-B, ViT-L, LLaMA, and GPT-2 models. We compare LoRA, NOLA, and SOLAR under adapter rank ($r = 4$). Tables 10 through 15 provide full parameter breakdowns. Byte-level analysis is presented in Table 13.

**ViT.**    For vision backbones, Table 10 and Table 11 report the number of representation parameters for query projections (Q) and classifier heads. In the experiments presented in the main paper, the classifier head parameters are excluded from comparison since they are identical across all methods following Koohpayegani et al. (2024). NOLA's parameter footprint for MLP projections is shown in Table 12 (following the setup in Koohpayegani et al. (2024)). Byte-level storage comparisons across quantization, used to produce Table 2 and Table 3 in the main paper, are provided in Table 13.

Table 10: Number of representation parameters for ViT-B (Rank = 4). Each row reports the parameter count for query projections and the classifier head using SOLAR and LoRA across different datasets. The classifier head parameter count is shared across methods and is computed as (`num_classes × 768 + num_classes`). For SOLAR, the query projection count corresponds to: number of layers $\times$ (top$_k$ coefficients for $A$ + top$_k$ coefficients for $B$ + encoded basis for $A$ + encoded basis for $B$) +1 (seed value). All SOLAR rows follow the form $N \rightarrow$ top$_k$ where $N$ is the original subspace size. For LoRA, the query projection count corresponds to: number of layers $\times$ (input dimension $\times$ rank for $A$ + rank $\times$ output dimension for $B$), where rank is 4.

| Method | Dataset | Query (Q) | Classifier Head |
|---|---|---|---|
| SOLAR | CIFAR-10 | $12 \times \left((1600 + 1600) + \frac{4000+4000}{32}\right) + 1 = 41{,}401$ | $10 \times 768 + 10 = 7{,}690$ |
| | CIFAR-100 | 41,401 | $100 \times 768 + 100 = 76{,}900$ |
| | Food-101 | 41,401 | $101 \times 768 + 101 = 77{,}669$ |
| | Tiny-ImageNet | 41,401 | $200 \times 768 + 200 = 154{,}000$ |
| LoRA | CIFAR-10 | $12 \times [(768 \times 4) + (4 \times 768)] = 73{,}728$ | $10 \times 768 + 10 = 7{,}690$ |
| | CIFAR-100 | 73,728 | $100 \times 768 + 100 = 76{,}900$ |
| | Food-101 | 73,728 | $101 \times 768 + 101 = 77{,}669$ |
| | Tiny-ImageNet | 73,728 | $200 \times 768 + 200 = 154{,}000$ |

**LLMs.**    For language models, parameter counts for adapter layers are detailed in Table 14 for LLaMA and in Table 15 for GPT-2 variants.

Table 11: Number of representation parameters for ViT-L (Rank = 4). Each row shows the parameter counts for Query projections and the classifier head using SOLAR and LoRA across different datasets. The classifier head parameter count is shared across methods and is calculated as (`num_classes × 1024 + num_classes`).

| Method | Dataset | Query (Q) | Classifier Head |
|---|---|---|---|
| SOLAR | CIFAR-10 | $24 \times \left((500 + 500) + \frac{1000 + 1000}{32}\right) + 1 = 25{,}501$ | $10 \times 1024 + 10 = 10{,}250$ |
| | CIFAR-100 | 25,501 | $100 \times 1024 + 100 = 102{,}500$ |
| | Food-101 | 25,501 | $101 \times 1024 + 101 = 103{,}625$ |
| | Tiny-ImageNet | 25,501 | $200 \times 1024 + 200 = 204{,}800$ |
| LoRA | CIFAR-10 | $24 \times [(1024 \times 4) + (4 \times 1024)] = 196{,}608$ | $10 \times 1024 + 10 = 10{,}250$ |
| | CIFAR-100 | 196,608 | $100 \times 1024 + 100 = 102{,}500$ |
| | Food-101 | 196,608 | $101 \times 1024 + 101 = 103{,}625$ |
| | Tiny-ImageNet | 196,608 | $200 \times 1024 + 200 = 204{,}800$ |

Table 12: Number of representation parameters for ViT-B (Rank = 4). Each row shows the parameter counts for MLP projections (for NOLA) and classifier head across datasets. The classifier head parameter count is shared across methods and is calculated as (`num_classes × 768 + num_classes`).

| Method | Dataset | MLP | Classifier Head |
|---|---|---|---|
| NOLA | CIFAR-10 | $12 \times 2 \times 2 \times 1000 + 1 = 48{,}001$ | $10 \times 768 + 10 = 7{,}690$ |
| | CIFAR-100 | 48,001 | $100 \times 768 + 100 = 76{,}900$ |
| | Food-101 | 48,001 | $101 \times 768 + 101 = 77{,}669$ |
| | Tiny-ImageNet | 48,001 | $200 \times 768 + 200 = 154{,}000$ |

# E  ADDITIONAL EXPERIMENTAL RESULTS

This section provides supplementary experimental results to further validate the claims made in the main paper. We present detailed performance metrics for additional model scales and include a crucial ablation study that compares SOLAR against a parameter-matched LoRA baseline.

## E.1  PERFORMANCE ON INTERMEDIATE-SCALE LLAMA MODELS

Table 16 extends our analysis to the LLaMA-3.2 3B and LLaMA-3.1 8B models, demonstrating SOLAR's consistent efficiency and performance on intermediate-scale architectures. The results show that SOLAR maintains the performance of the original LoRA adapters while achieving parameter reductions of over 90%.

## E.2  COMPRESSION OF ADAPTIVE-RANK PEFT METHODS (ADALORA)

To evaluate SOLAR on more recent PEFT methods, we applied it to AdaLoRA, which produces adaptive-rank adapter matrices ($\mathbf{A}$ and $\mathbf{B}$). SOLAR compresses these trained adapters post-hoc, using an initial rank of $r = 8$ and a target average rank of $r = 1$ on LLaMA-3.2 3B and LLaMA-2 13B. As shown in Table 17, SOLAR significantly reduces adapter parameters while preserving MMLU performance.

### E.2.1  EXPERIMENTS WITH 2-BIT QUANTIZATION

To further validate SOLAR's robustness to aggressive quantization, we conducted additional experiments with 2-bit quantization on LLaMA-2 13B and LLaMA-3.1 8B. The results, summarized in Table 18, confirm that SOLAR remains effective while drastically reducing parameter counts.

## E.3  EXTREME COMPRESSION

In this section, we report additional experiments demonstrating SOLAR's ability to achieve extreme compression while retaining competitive accuracy. These results complement the main paper by

Table 13: Byte-level footprint of representation parameters for ViT-B and ViT-L using LoRA and SOLAR. Each value reflects the total number of bytes required to store adapter updates (excluding classifier heads). For LoRA, storage is computed as: number of layers $\times$ (rank $\times$ output dimension for $B$ + input dimension $\times$ rank for $A$) $\times$ precision in bytes (e.g., 4 bytes for 32-bit float). For SOLAR, storage is computed as: number of layers $\times$ (top$_k$ coefficients for $A$ + top$_k$ coefficients for $B$ + encoded basis vectors for $A$ + encoded basis for $B$) $\times$ precision in bytes, plus 1 byte to store a random seed. For example, the row "$500 \rightarrow 50$" denotes that 500-dimensional subspaces are sparsified to top-$k = 50$ coefficients, with encoded bases represented at 1 bit per element (8 elements per byte).

| Method | Representation Footprint (Bytes) |
|---|---|
| LoRA ($r=1$) | $12 \times [(768 \times 1) + (1 \times 768)] \times 4 = 73{,}728$ |
| SOLAR for ViT-B 8Bit ($r=1, 500 \rightarrow 50$) | $12 \times \left[(50 + 50) + \frac{500}{8}\right] \times 1 + 1 = 1{,}951$ |
| SOLAR for ViT-B 8Bit ($r=1, 100 \rightarrow 10$) | $12 \times \left[(10 + 10) + \frac{100}{8}\right] \times 1 + 1 = 391$ |
| LoRA ($r=4$) | $24 \times [(1024 \times 4) + (4 \times 1024)] \times 4 = 786{,}432$ |
| SOLAR for ViT-L 32Bit ($r=4, 4000 \rightarrow 1600$) | $24 \times \left[(1600 + 1600) + \frac{4000}{32}\right] \times 4 + 1 = 319{,}201$ |
| SOLAR for ViT-L 16Bit ($r=4, 4000 \rightarrow 1600$) | $24 \times \left[(1600 + 1600) + \frac{4000}{16}\right] \times 2 + 1 = 165{,}601$ |
| SOLAR for ViT-L 8Bit ($r=4, 4000 \rightarrow 1600$) | $24 \times \left[(1600 + 1600) + \frac{4000}{8}\right] \times 1 + 1 = 88{,}801$ |
| SOLAR for ViT-L 4Bit ($r=4, 4000 \rightarrow 1600$) | $24 \times \left[(1600 + 1600) + \frac{4000}{4}\right] \times 0.5 + 1 = 50{,}401$ |

Table 14: Number of representation parameters for LLaMA-3 models using LoRA, NOLA, and SOLAR. Each row reports total adapter parameters for attention projections (Q and V for LoRA and NOLA; Q and K for SOLAR). Output heads and MLP layers are frozen. For LoRA, the parameter count is computed as: number of layers $\times$ (input dimension $\times$ rank for $B$ + rank $\times$ output dimension for $A$ +). Due to differing dimensions between $A$ and $B$ in LoRA, the table computes the contributions for Q and V projections separately. For NOLA, it is computed as: number of layers $\times$ 2 $\times$ (number of random basis vectors), assuming separate basis sets for $A$ and $B$. For SOLAR, the count is: number of layers $\times$ 2 $\times$ (top$_k$ coefficients for $B$ + top$_k$ for $A$ + encoded bases for $B$ + encoded bases for $A$), plus 1 byte to communicate or store the shared seed.

| Model (Rank) | Configuration | Total Parameters |
|---|---|---|
| LLaMA-3.2 1B ($r=8$) | 16 layers (Q, V) | $16 \times [(2048 \times 8 + 8 \times 2048) + (2048 \times 8 + 8 \times 512)] = 851{,}968$ |
| NOLA | 16 layers (Q, V) | $16 \times 2 \times (1000 + 1000) = 64{,}000$ |
| SOLAR ($r=8, 4K \rightarrow 1.2K$) | 16 layers (Q, V) | $16 \times 2 \times \left(1200 + 1200 + \frac{4000}{32}\right) + 1 = 80{,}801$ |
| LLaMA-3.2 3B ($r=1$) | 28 layers (Q, V) | $28 \times [(3072 \times 1 + 1 \times 3072) + (3072 \times 1 + 1 \times 1024)] = 286{,}720$ |
| NOLA | 28 layers (Q, V) | $28 \times 2 \times (1000 + 1000) = 112{,}000$ |
| SOLAR ($r=1, 1000 \rightarrow 150$) | 28 layers (Q, V) | $28 \times 2 \times \left(150 + 150 + \frac{1000}{32}\right) + 1 = 18{,}551$ |
| LLaMA-3.1 8B ($r=1$) | 32 layers (Q, V) | $32 \times [(4096 \times 1 + 1 \times 4096) + (4096 \times 1 + 1 \times 1024)] = 425{,}984$ |
| NOLA | 32 layers (Q, V) | $32 \times 2 \times (1000 + 1000) = 128{,}000$ |
| SOLAR ($r=1, 1000 \rightarrow 300$) | 32 layers (Q, V) | $32 \times 2 \times \left(300 + 300 + \frac{1000}{32}\right) + 1 = 40{,}401$ |

highlighting scenarios where communication and storage constraints are especially strict (e.g., distributed or on-device learning).

Table 19 shows evaluations on four vision datasets using ViT-B under different compression budgets. We quantify the bit-level representation footprint assuming 32-bit precision during training and apply 8-bit quantization to the SOLAR coefficients after top-$k$ selection. Compared to LoRA ($r = 1$), SOLAR reduces the adapter footprint by up to 99% (from 74KB to 0.4KB) with only minor drops in accuracy. These results illustrate that SOLAR enables fine-grained tradeoffs between accuracy and storage cost under extreme compression budgets.

Table 15: Number of trainable adapter parameters for GPT-2 models using LoRA, NOLA, and SOLAR. Each row reports the total number of parameters added to the query and value projections (Q and V). All configurations freeze the output heads and MLP layers. For LoRA, the parameter count is computed as: number of layers $\times 2 \times \big($input dimension $\times$ rank for $B$ + rank $\times$ output dimension for $A\big)$. For NOLA, the parameter count is: number of layers $\times 2 \times$ (number of random basis vectors), assuming separate basis sets for Q and V. For SOLAR, the parameter count is: number of layers $\times 2 \times \big($top$_k$ coefficients for $B$ + top$_k$ coefficients for $A$ + encoded bases for $B$ + encoded bases for $A\big)$, plus 1 for the shared seed.

| Model (Rank) | Configuration | Total Parameters |
|---|---|---|
| GPT-2 Small ($r$=4) | 12 layers (Q, V) | $12 \times 2 \times (768 \times 4 + 4 \times 768) = 147{,}456$ |
| NOLA | 12 layers (Q, V) | $12 \times 2 \times (1000 + 1000) = 48{,}000$ |
| SOLAR ($r$=1, $1000 \rightarrow 300$) | 12 layers (Q, V) | $12 \times 2 \times \big(300 + 300 + \frac{1000}{32}\big) + 1 = 15{,}150$ |
| SOLAR ($r$=1, $100 \rightarrow 90$) | 12 layers (Q, V) | $12 \times 2 \times \big(90 + 90 + \frac{100}{32}\big) + 1 = 4{,}396$ |
| GPT-2 Medium ($r$=4) | 24 layers (Q, V) | $24 \times 2 \times (1024 \times 4 + 4 \times 1024) = 393{,}216$ |
| NOLA | 24 layers (Q, V) | 350,000 Koohpayegani et al. (2024) |
| SOLAR ($r$=4, $1000 \rightarrow 300$) | 24 layers (Q, V) | $24 \times 2 \times \big(300 + 300 + \frac{1000}{32}\big) + 1 = 30{,}301$ |
| SOLAR ($r$=4, $100 \rightarrow 90$) | 24 layers (Q, V) | $24 \times 2 \times \big(90 + 90 + \frac{100}{32}\big) + 1 = 8{,}791$ |

Table 16: Model representation efficiency for LLaMA 3B and 8B models. For the 8B model, all methods use 4-bit quantization, making the LoRA baseline equivalent to QLoRA.

| Model | | LLaMA-3.2 3B | | | LLaMA-3.1 8B (4-bit) | |
|---|---|---|---|---|---|---|
| Method | LoRA $r$=1 | NOLA 1000 bases | SOLAR SOLAR$_{r=1(1K \rightarrow 0.1K)}$ | LoRA $r$=1 | NOLA 1000 bases | SOLAR SOLAR$_{r=1(1K \rightarrow 0.3K)}$ |
| # Params | 287K | 112K | **16K** (94% ↓) | 425K | 128K | **40K** (91% ↓) |
| Val Loss | **1.02** | 1.31 | **1.04** | **0.89** | 1.01 | **0.90** |
| MMLU Acc | **54.0** | 52.7 | **54.0** | **60.9** | 56.1 | **60.9** |

# F    SCALABILITY TO LARGER VISION MODELS

To validate that SOLAR remains effective and computationally tractable on larger-scale models, we conducted experiments on the ViT-G/14 architecture. This model is substantially larger than the ViT-B/L backbones used in our main experiments, providing a strong test of scalability.

We fine-tuned a ViT-G/14 model on the full CIFAR-10, CIFAR-100, Food-101, and T-ImageNet datasets using a LoRA adapter with rank $r = 4$. We then applied SOLAR with a basis pool of 8,000 vectors, selecting the top 4,000 coefficients to form the compressed adapter.

As shown in Table 20, SOLAR successfully preserves the performance of the original LoRA adapter with negligible accuracy drops, while reducing the adapter's parameter count by 31% (from 492K to 340K). This result demonstrates that SOLAR's core mechanisms—including SVD extraction and sparse reconstruction—scale effectively to larger models without sacrificing compression efficiency or task performance.

## F.1    ABLATION STUDY: BUDGET-MATCHED LORA COMPARISON

To further validate the efficiency of our compression strategy, we conduct an ablation study directly comparing SOLAR to a budget-matched LoRA baseline, as suggested by reviewer feedback.[1] This comparison is critical to demonstrate that SOLAR's benefits extend beyond mere parameter reduction and offer a more effective performance-compression trade-off than simply training a lower-rank adapter from scratch.

As shown in Table 21, fine-tuning a LoRA adapter with a reduced rank (r=2) to match the parameter count of the compressed SOLAR adapter results in a significant performance degradation across all

Table 17: SOLAR applied to AdaLoRA adapters on intermediate-scale LLaMA models.

| Method | # Params (Adapter) | MMLU Accuracy |
|---|---|---|
| AdaLoRA (Baseline, 3B) | 305K | 54.8% |
| SOLAR (on AdaLoRA, 3B) | **16K** | 54.7% |
| AdaLoRA (Baseline, 13B) | 871K | 57.9% |
| SOLAR (on AdaLoRA, 13B) | **16K** | 57.7% |

Table 18: 2-bit quantization experiments comparing LoRA (QLoRA) and SOLAR.

| Method | Quantization | # Params | MMLU Acc |
|---|---|---|---|
| LoRA (QLoRA) - LLaMA-2 13B | 2-bit | 410K | 53.1 |
| SOLAR$_{r=1(1K\rightarrow0.3K)}$ - LLaMA-2 13B | 2-bit | 51K | 53.1 |
| LoRA (QLoRA) - LLaMA-3.1 8B | 2-bit | 363K | 58.4 |
| SOLAR$_{r=1(1K\rightarrow0.3K)}$ - LLaMA-3.1 8B | 2-bit | 40K | 58.4 |

tasks. In contrast, SOLAR, when applied to the higher-performing LoRA (r=4) adapter, successfully preserves task accuracy while achieving a comparable parameter budget. This highlights that SOLAR retains the expressive power of the original higher-rank adapter, a feat not achievable by simply reducing the rank during training. All experiments were conducted on the full datasets using the ViT-B backbone, with results reported as the mean accuracy over five independent runs to ensure statistical robustness.

## G  COMPARISON WITH SIMPLE SVD TRUNCATION

To compare against simple post-hoc SVD truncation, we evaluate SOLAR's performance against SVD applied directly to the LoRA update $\Delta W$. Since the LoRA adapter $\Delta W$ already has rank $r$, SVD only provides compression if the truncation rank is set lower than $r$. We use an initial LoRA rank of $r = 4$ and truncate the SVD to rank 1. In contrast, SOLAR achieves a much smaller footprint by reparameterizing the update in the foundation model's subspace. The results are summarized in Table 22.

## H  APPLICATION TO FEDERATED LEARNING

One of the motivations for developing SOLAR is to reduce communication overhead in distributed learning scenarios, such as Federated Learning (FL). In typical FL setups, clients fine-tune a model on their local data and transmit the resulting model updates (e.g., LoRA adapters) to a central server for aggregation. As highlighted by recent work Mhanna & Assaad (2024), communication—not computation—is often the primary bottleneck. Transmitting full adapters from thousands of clients can generate enormous data transfer loads. For example, in an FL setup with 10,000 clients—1,000 participating in each of 10 training rounds—transmitting 74 KB LoRA adapters per client would amount to 740 GB of total data transfer.

SOLAR addresses this challenge as a lightweight, post-hoc compression utility. After local training, each client can compress its adapter with SOLAR before transmission. The server then receives only the sparse coefficients and a random seed, drastically reducing per-client communication costs.

To demonstrate SOLAR's effectiveness in distributed settings, we simulated a 10-client FL environment. We compare a baseline where clients transmit full LoRA adapters with a scenario where clients transmit SOLAR-compressed adapters. Each client fine-tunes a ViT-B model on CIFAR-10 with LoRA ($r = 4$), under two data distribution scenarios: an IID baseline and a non-IID distribution generated via a Dirichlet process with a concentration parameter of 0.5. The simulation runs for 30 communication rounds, with one epoch of local training per client per round.

As shown in Table 23, the performance gap between full LoRA adapters and SOLAR-compressed adapters is minimal in both IID and non-IID settings. This demonstrates that SOLAR's compression

Table 19: Evaluation of extreme compression on ViT-B. We report bit-level representation footprint (32-bit baseline) and top-1 accuracy over 5 runs. All models are trained for 10 epochs.

| Method | Byte Footprint | Oxford Pets | SUN397 | CUB-200 | ImageNet-1K |
|---|---|---|---|---|---|
| LoRA ($r$=1) | 74KB | $\mathbf{93.0}_{\pm 0.5}$ | $\mathbf{74.3}_{\pm 0.3}$ | $\mathbf{84.7}_{\pm 0.4}$ | $\mathbf{81.5}_{\pm 0.6}$ |
| SOLAR ($r$=1, $500 \rightarrow 50$) | $\underline{2\text{KB}}$ (97% ↓) | $\underline{91.2}_{\pm 0.6}$ | $\underline{72.4}_{\pm 0.4}$ | $\underline{81.4}_{\pm 0.5}$ | $\underline{80.7}_{\pm 0.4}$ |
| SOLAR ($r$=1, $100 \rightarrow 10$) | $\mathbf{0.4\text{KB}}$ (99% ↓) | $90.3_{\pm 0.7}$ | $72.4_{\pm 0.5}$ | $81.3_{\pm 0.6}$ | $80.6_{\pm 0.5}$ |

Table 20: Scalability of SOLAR on the ViT-G/14 model. Results show top-1 accuracy (%) on full datasets.

| Method | # Params | CIFAR-10 | CIFAR-100 | Food-101 | T-ImageNet |
|---|---|---|---|---|---|
| LoRA ($r = 4$) | 492K | 99.4 | 94.6 | 91.2 | 92.8 |
| SOLAR ($r = 4, 8\text{K} \rightarrow 4\text{K}$) | 340K (31% ↓) | 99.4 | 94.5 | 91.2 | 92.8 |

Table 21: Comparison of SOLAR with a budget-matched LoRA (r=2) baseline on ViT-B. While LoRA (r=2) has a similar parameter count to the compressed SOLAR adapter, it shows a clear performance degradation. SOLAR maintains performance comparable to the original, higher-rank LoRA (r=4).

| Method | #Params | CIFAR-10 | CIFAR-100 | Food-101 | T-ImageNet |
|---|---|---|---|---|---|
| LoRA ($r = 4$) | 74K | 98.3 | 90.3 | 87.6 | 88.8 |
| LoRA ($r = 2$) | 37K | 97.1 | 89.0 | 85.5 | 87.4 |
| SOLAR ($r = 4, 4\text{K} \rightarrow 1.6\text{K}$) | 41K | 98.3 | 89.8 | 87.0 | 87.9 |
| SOLAR ($r = 4, 4\text{K} \rightarrow 0.8\text{K}$) | 22K | 97.0 | 89.0 | 85.2 | 87.4 |

Table 22: Comparison of SOLAR and simple SVD truncation against standard LoRA adapters on multiple vision datasets. The table reports classification accuracy and the corresponding byte footprint of the adapter parameters after compression. SOLAR consistently reduces the parameter size while preserving or improving performance.

| Method | Byte Footprint | Oxford Pets | SUN397 | CUB-200 | ImageNet-1K |
|---|---|---|---|---|---|
| LoRA ($r = 1$) | 74KB | 93.0 | 74.3 | 84.7 | 81.5 |
| LoRA ($r = 4$) | 297KB | 94.2 | 75.6 | 86.0 | 82.8 |
| SVD truncation on LoRA | 74KB | 92.7 | 73.3 | 83.6 | 80.8 |
| SOLAR on LoRA ($r = 1$) | 8KB | 92.6 | 73.9 | 84.2 | 81.3 |
| SOLAR on LoRA ($r = 4$) | 8KB | 93.9 | 75.0 | 85.4 | 82.4 |

does not disproportionately harm aggregation performance, even under significant data heterogeneity. Our experiment confirms that SOLAR can serve as a post-training, plug-and-play module to reduce communication costs in standard FL frameworks without requiring complex changes to the aggregation strategy.

Table 23: Performance of SOLAR on ViT-B under IID and non-IID data distributions in a simulated 10-client federated learning environment.

| Method | # Params | CIFAR-10 (IID) | CIFAR-10 (non-IID) |
|---|---|---|---|
| LoRA ($r = 4$) | 74K | 93.7 | 87.4 |
| SOLAR ($r = 4, 4\text{K} \rightarrow 2\text{K}$) | 51K (31% ↓) | 93.2 | 86.7 |

