# OpenReview forum: "SOLAR: Communication-Efficient Model Adaptation via Subspace-Oriented Reparametrization"
_ICLR.cc/2026/Conference — Submitted to ICLR 2026_

### Official Review · Reviewer_qfT9 · 2025-10-28

**Soundness:** 3
**Presentation:** 3
**Contribution:** 3
**Rating:** 6
**Confidence:** 4

**Summary:**

This paper proposes the SOLAR method, which constrains parameter updates to the subspace spanned by singular vectors through subspace-oriented reparameterization and combines this with a sparse composition mechanism, achieving significant reductions in communication and storage costs. The paper also provides a theoretical upper bound on the reconstruction error and demonstrates the method’s effectiveness on NLP and CV tasks.

**Strengths:**

1. The paper introduces a subspace-oriented reparameterization approach that differs from mainstream low-rank methods.  By constraining parameter updates to the subspace spanned by singular vectors, it offers a novel perspective for PEFT research.
2. The combination of the reparameterization strategy with a sparse composition mechanism is well-motivated and supported by solid theoretical grounding.
3. The paper is reasonably well-structured, with the method presented in a mathematically rigorous way and the experimental tables and results clearly organized.

**Weaknesses:**

1. While the paper introduces a theoretical upper bound on the reconstruction error, the connection between this bound and the observed empirical performance could be discussed in more depth.
2. The evaluation might be further strengthened by including comparisons with some of the most recent efficient PEFT methods.

**Questions:**

The paper presents a theoretical upper bound on the reconstruction error, but the discussion of its tightness and its relation to experimental results is not sufficiently developed. Including experiments that compare the theoretical bound with the observed reconstruction error would help strengthen the theoretical contribution.  In addition, the empirical evaluation could be further improved by incorporating comparisons with more recent efficient PEFT methods to enhance the overall persuasiveness of the work.

---

> ### Author Response · Authors · 2025-11-21
>
> **W1** and **Q1**: We thank the reviewer for raising these points. The connection between our theoretical bound (Theorem 1) and empirical results is the focus of Section 3.4 and Figure 3. Theorem 1 states that the compression error ($C_2$) is bounded by terms related to the basis pool size ($N$) and the sparsity budget ($k$). Specifically, $C_2$ (the total compression error) can be minimized by enlarging the basis pool $N$ and increasing the sparsity budget $k$. Figure 3 provides direct empirical validation of this: the error ("Accuracy Difference" on the y-axis) decreases as the budget $k$ (x-axis) increases, and for any fixed $k$, a larger basis pool $N$ (color scale) further reduces the error.
> This experiment directly links our theory's mechanisms to the observed empirical trade-offs, confirming that $N$ and $k$ are effective levers for controlling the reconstruction quality. We will revise the discussion in Section 3.4 to make this connection to Theorem 1 more explicit in the final version.
>
>
> **W2** and **Q2**: To strengthen our experimental evaluation and include more recent PEFT methods, we will add AdaLoRA to our baselines and update Tables 1, 5, and 16 in the revised version. AdaLoRA produces a final set of optimized, adaptive-rank adapter matrices ($\mathbf{A}$ and $\mathbf{B}$). SOLAR then takes these final matrices directly as input and applies our standard post-hoc compression procedure.
>
> To empirically validate this, we conducted new experiments where SOLAR was applied to a trained AdaLoRA adapter on the LLaMA-3.2 3B and  LLaMA-2 13B models. Our preliminary results show that SOLAR can successfully compress the AdaLoRA adapter while preserving MMLU performance, demonstrating that SOLAR is model-agnostic and fully compatible with more sophisticated PEFT structures. For AdaLoRA, the initial rank is set to $r=8$ and the final target average rank is set to $r=1$. All other experimental settings follow Section 3.2.
>
>
> | Method               | # Params (Adapter)    | MMLU Accuracy |
> | :------------------- | :-------------------- | :------------- |
> | AdaLoRA (Baseline, 3B)   | 305K                  | 54.8%          |
> | SOLAR (on AdaLoRA, 3B)   | 16K           | 54.7%          |
> | AdaLoRA (Baseline, 13B)   | 871K                  | 57.9%          |
> | SOLAR (on AdaLoRA, 13B)   | 16K          | 57.7%          |

---

### Official Review · Reviewer_BHU1 · 2025-10-30

**Soundness:** 2
**Presentation:** 1
**Contribution:** 2
**Rating:** 2
**Confidence:** 5

**Summary:**

This paper introduces SOLAR, a parameter-efficient finetuning (PEFT) method that aims to reduce communication costs, i.e. the number of tunable parameters. The key idea is to represent the weight update as a linear combination of multiple basis matrices, where each basis is generated by randomly permuting the original pretrained weights. The only learnable parameters are the coefficients of this linear combination, effectively restricting finetuning to a low-dimensional subspace. SOLAR is evaluated on a range of model architectures, including LLaMA, GPT, and ViT.

**Strengths:**

The idea of reparameterizing finetuning in a low-dimensional subspace is conceptually interesting and has potential implications for lightweight adaptation.

The approach is simple and easy to understand, which makes it appealing for practical deployment scenarios.

**Weaknesses:**

1. Although the number of learnable parameters is small, the subspace matrices $M_A$ and $M_B$ themselves occupy non-trivial memory. This undermines the claim of efficiency, as these auxiliary tensors could impose significant additional storage and communication overhead.

2. The method implicitly assumes that the task-specific update $\Delta W$ lies close to a subspace spanned by random permutations of the pretrained weights $W$. This is a strong and non-obvious assumption. The paper should provide empirical evidence or theoretical justification to support the existence of such subspace similarity.

3. Unrepresentative experimental setup:
- The experimental design raises concerns about the generality of the results. Using only 10 samples for finetuning is not representative of typical adaptation scenarios.
- The reliance on outdated models (e.g., GPT-2) and small-scale datasets (CIFAR-10/100) limits the relevance and impact of the findings.

4. The reported gains are significant — as shown in Table 1, SOLAR does NOT outperform baselines such as LoRA. In addition, ablation studies on the construction of the subspace (e.g., number of bases, permutation strategy) are missing, leaving the effectiveness of key design choices unclear.

5. The writing is not well structured and organized. It takes extra efforts to understand some unexplained concepts, like subspace similarity, the alignment of principal directions, PEFT structure, etc.

**Questions:**

See weakness

---

> ### Author Response · Authors · 2025-11-21
>
> **W1**: We thank the reviewer for raising these points. Regarding the transmission/storage cost of the basis matrices ($M_A, M_B$), we would like to clarify that these bases (auxiliary tensors) are never stored or transmitted. As described in Section 2.2 and clearly illustrated in Figure 1, the bases are not pre-computed or kept in memory. Instead, they are generated deterministically on-the-fly using a Pseudo-random generator. The key component is that the randomized basis pool can be fully reconstructed on any device using only a single shared random seed and the readily available weights of the foundation model ($W$). Thus, because the bases are fully reproducible from this single seed, SOLAR does not incur the cost of storing or communicating any dense basis tensors. The minimal payload required to represent a compressed adapter consists only of: (i) a single random seed to regenerate the basis vectors, (ii) an encoded list of selected basis indices, and (iii) the small set of selected sparse coefficients ($\mathbf{\alpha, \beta}$). This approach yields an extremely small data footprint. Thus, SOLAR’s payload is extremely small: For example, on the LLaMA-3.2 13B model, this mechanism reduces the adapter's parameter count by up to 94% (from 819K parameters for LoRA to 51K parameters for SOLAR), while maintaining the performance of the original LoRA adapter.
>
> **W2**: We would like to clarify that the paper does not assume that task-specific updates lie in a subspace spanned by random permutations of pretrained weights. In fact, SOLAR is explicitly motivated and designed based on the assumption that the adapter update $\Delta W$ and the foundation model $W$ share an alignment in their principal directions. This alignment is theoretically supported by frameworks such as Neural Tangent Kernel (NTK) theory, which suggests that updates concentrate along the low-curvature subspaces of $W$. We provide direct empirical evidence of this phenomenon in Section 3.4 ("Subspace Analysis"). Figure 2 formally quantifies the overlap in the principal singular vector subspaces of $W$ and $\Delta W$, confirming that the fine-tuned model emphasizes directions already present in the foundation model.
>
> This similarity motivates our core design: we build the basis pool by perturbing $W$'s singular vectors, thus creating a set of bases aligned with $W$'s subspace, rather than relying solely on unstructured random bases (as in the NOLA paper). As Tables 1 and 5 show, SOLAR requires substantially fewer parameters than other baselines to achieve comparable accuracy.
>
> **W3**: The few-shot 10-sample configuration is only one of the evaluation regimes used for ViT models; as clearly described in Section 3.1 and Table 1, all vision experiments are conducted under **both** few-shot (10 samples per class) and full-data settings. Moreover, our experimental scope extends far beyond GPT-2 or CIFAR-10/100. While GPT-2 is included for completeness and comparability with prior PEFT work, the core of our evaluation focuses on modern, large-scale foundation models, including LLaMA-3.2 1B, LLaMA-3.2 3B, LLaMA-3.1 8B, and LLaMA-2 13B. On the vision side, we evaluate ViT-B and ViT-L in the main paper, and additionally scale to ViT-G/14 in the appendix. Our datasets likewise go beyond CIFAR: we report results on ImageNet-1K, Food-101, Tiny-ImageNet, the Alpaca instruction-tuning corpus, and the MMLU benchmark to assess generalization. Collectively, our experiments span models from 1B to 13B+ parameters and include standard, large-scale benchmarks in both vision and language, evaluated under few-shot and full-data training regimes. We therefore believe the experimental design is broad, modern, and fully representative of typical adaptation scenarios, and supports the generality and practical relevance of SOLAR.

---

> > ### Author Response · Authors · 2025-11-21
> >
> > **W4**. We would like to clarify that the goal of our paper is to match LoRA's performance while drastically reducing communication and storage costs.
> > SOLAR is designed as a post-training compression method, and our results consistently show that it meets this objective. Across models and modalities, SOLAR preserves LoRA performance while dramatically reducing adapter size. For example, on ViT-B (Table 1), SOLAR matches LoRA’s full-data accuracy on CIFAR-10 (98.3%) while using 45% fewer parameters (41K vs. 74K). On LLaMA-2 13B (Table 5), SOLAR matches LoRA’s MMLU score (54.5) with a 94% reduction in adapter size (819K → 51K). Similarly, on GPT-2 Small (Table 6), SOLAR preserves LoRA’s METEOR score (29.7) while achieving a 90% compression ratio (147K → 15K). These results demonstrate that SOLAR reliably maintains performance under substantial parameter reduction.
> > We respectfully disagree with the reviewer's claim that this ablation is missing. The entire focus of Section 3.4 and Figure 3 is an ablation study. In this section, we "analyze the effect of varying the basis pool size (N) and the number of selected top_k components (k)", which are the key design choices of SOLAR. Figure 3 provides the direct empirical validation, clearly showing that performance (y-axis) improves with a larger budget $k$ (x-axis) and a larger basis pool $N$ (color scale). Regarding the permutation strategy, we clarify that our subspace construction does not leverage any permutation strategy; instead, we randomly generate $N$ bases and select the top_k components to maintain the performance while reducing the number of parameters.
> >
> >
> > **W5**: We respectfully believe the paper follows a clear, logical structure (Section 1: Introduction, Section 2: Method, Section 3: Experiments, Section 4: Related Works). Nevertheless, we have polished the writing further to improve the readability. Regarding the specific concepts mentioned, we believe they are defined and explained in the text.
> >
> > We explicitly define "Subspace Similarity" and "Alignment" as synonymous. This is first stated in the Abstract ("...exploiting the subspace similarity (the alignment of principal directions)...") and elaborated upon in Section 2.2 ("This alignment (i.e., the overlap in the principal directions of W and $\Delta W$)..."). A full, dedicated discussion with a formal quantification is then provided in Section 3.4 ("Subspace Analysis") and Figure 2, where we explicitly define the similarity function $\phi(W,\Delta W,i,j)$ to measure this alignment empirically.
> >
> > PEFT Structure is introduced in the Introduction (e.g., LoRA) and then provided with a precise mathematical formulation in Section 2.1, which defines the LoRA structure as $\Delta W=BA$.

---

### Official Review · Reviewer_Z2Yv · 2025-11-01

**Soundness:** 3
**Presentation:** 3
**Contribution:** 3
**Rating:** 6
**Confidence:** 4

**Summary:**

In my understanding, SOLAR is a post-training compression method designed to make parameter-efficient fine-tuning (PEFT) adapters like LoRA much lighter for communication and storage. It uses the sparsity of LoRA to its advantage. Instead of storing full adapter matrices, SOLAR converts them into sparse combinations of basis vectors derived from the foundation model’s singular vectors, each with a slight random perturbation. This approach leverages the natural alignment between fine-tuned updates and the model’s subspace, allowing adapters to be reconstructed using only a few coefficients and a random seed. As a result, SOLAR can shrink adapter size by a large margin while preserving accuracy, making it ideal for distributed or resource-constrained environments without changing the original training process.

**Strengths:**

Well-Motivated Idea: The paper addresses a critical bottleneck in PEFT, which is the storage overhead. This can even be discussed more in the paper, as for mobile applications which might leverage lots of LoRA on-device, it is super useful to save space. The authors leverage LoRAs sparsity and subspace alignment for compression.

Theoretical Backing: The paper is built on a strong theoretical foundation, from the subspace analysis(Amplification factor) literature on LoRA. The authors show formal reconstruction error bounds and explain why subspace-oriented bases preserve task performance.

Empirical: The method impressive results across vision and language tasks, achieving up to 98 to 99% adapter size reduction with negligible accuracy loss. Robustness under quantization and scalability to large models further validate the approach.

Significance: Offers a practical solution for storing and transmitting large numbers of LoRA adapters on-device or in federated learning setups, making PEFT more viable in real-world deployments.

**Weaknesses:**

[W1] Not sure if I missed it, but I did not find a comparison of  subspace-based compression v/s a simple post-hoc compression approach such as [1] (Simply dropping singular values)

[W2] There is a table on effect of quantization of SOLAR, but quantization itself if a method of saving memory footprint. There needs to be a comparison of how SOLAR performs v/s quantization (LoRA+different quantization levels v/s SOLAR+different quantization levels)

[W3] I understand a case when SOLAR performs well in compressing LoRA weights when the base model itself can represent what the LoRA has learnt (there is a method to extract this information from the subspace of the model). However, many adapters are highly orthogonal (and are trained to be highly orthogonal) to the model weights. This is also analogous to the information present in LoRA being part of nullspace of the model (discussed in [2]). The authors must analyze the effect of SOLAR compression on tasks with varying range of orthogonality (increasing amp factor to the model weights).

[1] https://arxiv.org/pdf/2509.10971
[2] https://arxiv.org/abs/2506.04244

**Questions:**

Q1. How does the method compress adaptive-rank selective PEFT methods (AdaLoRA, SORA, FouRA)? Will the memory saved be lesser?
Q2. Methods such as FouRA and Wavelet-based LoRAs try to demonstrate that singular values are not widely spread out in the low rank subspace. How would the subspace-based compression affect these models?

---

> ### Author Response · Authors · 2025-11-21
>
> **W1**: We thank the reviewer for raising these points. A simple post-hoc SVD truncation of the adapter weights ($\Delta W = BA$) as discussed in [1] (PHLoRA) is indeed a natural baseline for post-training compression, and we will include it in the final version. In particular, the following results are obtained by applying SVD to $\Delta W$ alone to expand Table 2 of the paper.
> PHLoRA extracts the update matrix by computing $\Delta W = W_{\text{ft}} - W_{\text{base}}$ and then applying a rank-$r$ truncated SVD to obtain the best rank-$r$ approximation of this delta. In our experiments, however, the fine-tuned model is obtained via LoRA, meaning $\Delta W$ already has rank at most $r_{\text{LoRA}}$. Consequently, applying PHLoRA with a rank equal to $r_{\text{LoRA}}$ yields an identical $\Delta W$ but provides no additional compression. The only scenario in which PHLoRA can further reduce size is when its truncated SVD uses a rank lower than $r_{\text{LoRA}}$. To illustrate this, we repeated the analysis in Table 2 using LoRA with rank 4, extracted the resulting $\Delta W$, and then applied PHLoRA with rank 1. We also emphasize that when the underlying fine-tuning already uses LoRA, PHLoRA cannot produce an adapter smaller than the lowest possible LoRA rank (i.e., rank 1).
>
> | Method | Byte Footprint | Oxford Pets | SUN397 | CUB-200 | ImageNet-1K |
> | :--- | :--- | :--- | :--- | :--- | :--- |
>  | LORA $(r=1)$ | 74KB | 93.0 | 74.3 | 84.7 | 81.5 |
>  | LORA $(r=4)$ | 297KB | 94.2 | 75.6 | 86.0 | 82.8 |
>  | Post-hoc SVD on LORA $(r=4)$  (PHLoRA $(r=1)$)|  74KB | 92.7 | 73.3 | 83.6 | 80.8 |
>  | SOLAR on LORA $(r=1)$ | 8KB | 92.6 | 73.9 | 84.2 | 81.3 |
>  | SOLAR on LORA $(r=4)$ | 8KB | 93.9 | 75.0 | 85.4 | 82.4 |
>
>
>
>
> To clarify further, we would like to emphasize one conceptual difference. A simple SVD truncation compresses $\Delta W$ using its own subspace. In contrast, SOLAR is motivated by "subspace similarity": the hypothesis that $\Delta W$ is better compressed in the subspace of the foundation model's weights, $W$. This approach is highly communication-efficient because the foundation model $W$ (and thus its SVD subspace) is already shared in the system. To reconstruct multiple $\Delta W$ adapters, each only needs to transmit/store the tiny set of sparse coefficients ($\alpha, \beta$) and a single random seed, as the basis is generated from the shared $W$.
>
> Moreover, a rank-1 SVD of $\Delta W$ is parametrically equivalent to a LoRA $(r=1)$ adapter, offering no additional compression. This highlights the need for a method like SOLAR, which fundamentally changes the representation to compress far beyond what a standard rank-1 decomposition can provide.
>
> **W2**: Quantization is indeed a compression technique, and SOLAR is designed to operate orthogonally to it, serving as a complementary post-training method. We would like to clarify that the requested comparison (LoRA+quant vs. SOLAR+quant) is already present in our main experiments. As stated in Table 5 (LLaMA-2 13B), our experiment already directly compares 4-bit QLoRA to 4-bit SOLAR. The results show SOLAR matches LoRA's MMLU accuracy (54.5) while reducing parameters by 94% (819K vs. 51K). A second, fully quantized comparison is also provided in Appendix E.1 (Table 16) for the LLaMA-3.1 8B model. Table 3 serves a different purpose: it isolates SOLAR to demonstrate its robustness to quantization, showing performance is stable even down to 4-bits.
> To further strengthen this, we will add experiments with more aggressive 2-bit quantization on LLaMA-2 13B and LLaMA-3.1 8B to the final paper. The preliminary results below confirm SOLAR's complementary benefits.
>
> | Method | Quantization | # Params | MMLU Acc |
> | :--- | :--- | :--- | :--- |
>  | LORA (QLORA) - LLaMA-2 13B | 2-bit | 410K | 53.1 |
> | SOLAR $(r=1, 1K \rightarrow 0.3K)$ - LLaMA-2 13B | 2-bit | 51K | 53.1 |
>  | LORA (QLORA) - LLaMA-3.1 8B | 2-bit | 363K | 58.4 |
> | SOLAR $(r=1, 1K \rightarrow 0.3K)$ - LLaMA-3.1 8B | 2-bit | 40K | 58.4 |

---

> > ### Author Response · Authors · 2025-11-21
> >
> > **W3**: SOLAR is explicitly motivated and designed for the case where the adapter update $\Delta W$ and the foundation model weights $W$ share subspace similarity. We provide strong empirical evidence for this alignment in Section 3.4 (Figure 2) and cite the theoretical basis for it (e.g., NTK theory). If a task required an adapter $\Delta W$ that is highly orthogonal to the $W$ subspace (i.e., it learns information purely in the model's nullspace), maintaining SOLAR's performance might require increasing the basis pool size ($N_A, N_B$).
> > The reviewer's cited work [2] (ProLoRA) provides evidence for this scenario. That paper decomposes the LoRA update into ($\Delta W_{s, \parallel}$) and nullspace ($\Delta W_{s, \perp}$) components. Their study shows that ignoring the nullspace results in deficient style transfer and a worse metric.
> > While exploring nullspace-aware basis construction is an important direction for future work, our method includes a partial mitigation for this. In Section 2.2.1, we note that our basis is "subspace-oriented randomized." We add controlled random perturbations ($\epsilon$) specifically to enrich this subspace, allowing it to modestly capture directions that are not perfectly aligned with the original principal components.
> >
> > **Q1**: We thank the reviewer for raising these questions. To strengthen our experimental evaluation and include more recent PEFT methods, we will add AdaLoRA to our baselines and update Tables 1, 5, and 16 in the revised version. AdaLoRA produces a final set of optimized, adaptive-rank adapter matrices ($\mathbf{A}$ and $\mathbf{B}$). SOLAR then takes these final matrices directly as input and applies our standard post-hoc compression procedure.
> >
> > To empirically validate this, we conducted new experiments where SOLAR was applied to a trained AdaLoRA adapter on the LLaMA-3.2 3B and  LLaMA-2 13B models. Our preliminary results show that SOLAR can successfully compress the AdaLoRA adapter while preserving MMLU performance, demonstrating that SOLAR is model-agnostic and fully compatible with more sophisticated PEFT structures. For AdaLoRA, the initial rank is set to $r=8$ and the final target average rank is set to $r=1$. All other experimental settings follow Section 3.2.
> >
> >
> >
> > | Method               | # Params (Adapter)    | MMLU Accuracy |
> > | :------------------- | :-------------------- | :------------- |
> > | AdaLoRA (Baseline, 3B)   | 305K                  | 54.8%          |
> > | SOLAR (on AdaLoRA, 3B)   | 16K           | 54.7%          |
> > | AdaLoRA (Baseline, 13B)   | 871K                  | 57.9%          |
> > | SOLAR (on AdaLoRA, 13B)   | 16K          | 57.7%          |
> >
> > **Q2**:  FouRA and wavelet-based LoRAs (WaRA) operate in transformed domains (Fourier or wavelet), but after training, they still produce a standard LoRA-style update $\Delta W = BA$ in the model’s weight space. SOLAR does not make assumptions about the singular value distribution of $\Delta W$. Instead, its core assumption is subspace similarity: that an effective PEFT update $\Delta W$, regardless of how it was obtained, tends to lie predominantly within the dominant singular subspace of the foundation model $W$. Under this assumption, SOLAR’s compression mechanism, projecting $\Delta W$ onto basis vectors derived from the SVD of $W$, remains effective.
> >
> > Moreover, even if transformed-domain methods such as FouRA or WaRA introduce components that are not perfectly aligned with $W$’s principal directions, SOLAR’s basis is not purely based on SVD. As described in Section 2.2.1, our basis is "subspace-oriented randomized": we add controlled Gaussian perturbations ($\epsilon$) to the singular vectors to enrich the basis pool. These perturbations help SOLAR capture components of $\Delta W$ that deviate from the dominant SVD directions, making the representation more robust to such mismatches.

---

### Official Review · Reviewer_jmkV · 2025-11-01

**Soundness:** 3
**Presentation:** 3
**Contribution:** 3
**Rating:** 4
**Confidence:** 3

**Summary:**

The author introduces the methods SOLAR (Subspace-Oriented Latent Adapter Reparameterization), a post-training compression method, which aims to target the communication and storage bottlenecks of PEFT method. It re-expresses the PEFT update in a more compact way, formulates each PEFT update as a linear combination of basis vectors formed from the foundation model’s singular vectors with controlled random perturbations. By exploiting subspace similarity, SOLAR remains compact and expressive with bounded reconstruction error. SOLRA offers interesting applications for deploying PEFT in distributed systems and edge devices.

**Strengths:**

- the results in Table 5 and Table 6 are quite impressive, especially because language models are the most scaled models, this huge reduction in parameter count can have significant implications for PEFT methods's communication and storage when applied to large language models
- interesting results provided in the subspace analysis, I think it greatly justifies the different alignments used in SOLAR compared to NOLA
- theoretical analysis of the SOLAR reconstruction error bound

**Weaknesses:**

- the author mentions in the abstract that it's method is model-agnositc and compatible with existing PEFT methods other than LoRA, while the authors only applies SOLAR to low-rank based methods. I think it is required to see whether I will be interested into seeing the results of SOLAR applied to PEFT methods with other examples, for example, the authors mentioned orthogonal finetuning methods in Section 2.0, I would interested to see if it also works there, because different than low-rank based methods, orthogonal matrices are full-rank, is the mentioned improvement of paramter-efficiency still present, also under lower model precision? I would be interested in additional results in general, at least results of Table 5 to add the results of OFT and quantized OFT (to the best of my knowledge, I think orthogonal finetuning also supports low-precision base model) and the reduction in parameters in llama and gpt are most significant. I think this will definitely strengthen the paper and as it shows SOLAR is generic and applicable to different sorts of PEFT methods

**Questions:**

- I am wondering will SOLAR also works with fine-tuning diffusion models? LoRA has often been applied into finetuning diffusion models, will the reparameterization affect the image generation quality?
- The reductions in parameters in ViT not seem be as significant as Llama and GPT,  what is the implication for that?
- I might miss something, but I am a bit confused with what SOLAR stores and how to calculate the trainable parameters of SOLAR: with a_iM_A and b_jM_B? (like the equation 5?)

---

> ### Author Response · Authors · 2025-11-21
>
> **W**: We thank the reviewer for raising this point. SOLAR is a post-hoc compression method that can be used in a plug-and-play manner with existing PEFT techniques. SOLAR is explicitly motivated and designed based on the assumption that the adapter update $\Delta W$ and the foundation model $W$ share an alignment in their principal directions. This alignment is theoretically supported by frameworks such as Neural Tangent Kernel (NTK) theory, which suggests that updates concentrate along the low-curvature subspaces of $W$. We provide direct empirical evidence of this phenomenon in Section 3.4 ("Subspace Analysis"). Figure 2 formally quantifies the overlap in the principal singular vector subspaces of $W$ and $\Delta W$, confirming that the fine-tuned model emphasizes directions already present in the foundation model. Therefore, we claim SOLAR is model-agnositc and compatible with existing PEFT methods.
>
> SOLAR compatibility is not restricted to low-rank adapters. Since OFT/BOFT defines fine-tuned weights as $W' = RW$, the effective adapter is $\Delta W = (R - I)W$. SOLAR applies directly to $\Delta W$, regardless of its rank, because $\Delta W$ inherits the principal directions of $W$. Thus, the OFT/BOFT update can be treated as input to SOLAR. For example, based on Table 3 of the OFT paper, OFT (b=16) achieves ~50% reduction in parameters compared to LoRA (r=32) on Llama-2-7B. However, the final storage size for the trained OFT adapter remains significantly higher than that of SOLAR, which achieves more than 90% reduction in the number of LoRA parameters.
>
> To strengthen our experimental evaluation, we conducted new experiments where SOLAR was applied to a trained AdaLoRA adapter on the LLaMA-3.2 3B and  LLaMA-2 13B.
> Our preliminary results show that SOLAR can compress the AdaLoRA while preserving MMLU performance. For AdaLoRA, the initial rank is set to $r=8$ and the final target average rank is set to $r=1$. All other experimental settings follow Section 3.2.
>
> | Method               | # Params (Adapter)    | MMLU Accuracy |
> | :------------------- | :-------------------- | :------------- |
> | AdaLoRA (Baseline, 3B)   | 305K                  | 54.8%          |
> | SOLAR (on AdaLoRA, 3B)   | 16K           | 54.7%          |
> | AdaLoRA (Baseline, 13B)   | 871K                  | 57.9%          |
> | SOLAR (on AdaLoRA, 13B)   | 16K          | 57.7%          |
>
> We will add AdaLoRA into our baselines and update Tables 1, 5, and 16 in the revised version.
>
> **Q1**: We thank the reviewer for raising these questions. Theoretically, SOLAR should be fully compatible. Our method is a model-agnostic compression framework that operates on the final trained LoRA adapters. Because it is not tied to the underlying model architecture (e.g., Transformer vs. U-Net), SOLAR can be applied to any model fine-tuned with LoRA-style adapters.
> Regarding the effect on image generation quality, we must be clear: as we state in our "Limitations and Future Work" section, SOLAR's effectiveness on "other modalities... or multimodal data" remains untested. While we hypothesize the same performance-versus-size trade-offs, this is a valuable direction for future work.
>
> **Q2**: The percentage reduction for LLMs is far higher (e.g., 90-94% for LLaMA/GPT) than for ViT (e.g., 45% for ViT-B). This is not because SOLAR is less effective on ViT; rather, it is because the baseline LoRA adapters for LLaMA and GPT are dramatically larger to begin with. For example, on LLaMA-2 13B, we compress from 819K parameters down to 51K, while for ViT-B ($r=4$), we compress from 74K down to 41K.
>
> The LLM adapters are much larger due to fundamental architectural differences. For instance, the LLaMA-3.1 8B model has 32 layers and a hidden dimension of 4096, with adapters placed in both the Query and Value projections, resulting in a 425K-parameter LoRA (r = 1) adapter. In contrast, our ViT-B baseline has only 12 layers, a dimension of 768, and adapters are placed only in the Query projection, resulting in a much smaller 74K parameter LoRA (r=4) adapter.
>
> **Q3**: The basis matrices (M_A​, M_B​) are never stored or transmitted. As shown in Figure 1, these bases are not stored tensors but are generated on-the-fly by a "Pseudo-random generator" using only a single shared seed.
> Therefore, the parameters SOLAR stores are not the dense M_A​ or M_B​ matrices. As detailed in Section 3.1, the only information stored or communicated consists of three components: a single random seed (to deterministically regenerate all basis vectors), an encoded list of selected basis indices (to know which top−k bases were used), and the sparse coefficients (α,β) for those selected indices.
> We provide the exact calculation for this parameter count in the appendix. The clearest example is in the caption for Table 10, which defines the SOLAR parameter count as: number of layers * (top_k coefficients for A + top_k coefficients for B + encoded basis for A + encoded basis for B) + 1 (seed value).

---

### Meta-Review · Area_Chair_5aPq · 2026-01-06

**Summary:**

SOLAR is a post-training compression method that reduces the communication and storage footprint of PEFT adapters such as LoRA. Building on NoLA (ICLR 2024), it reparameterizes the LoRA update factors as linear combinations of pseudo-random basis matrices, so an adapter can be represented by a shared random seed and a small set of combination coefficients. The key novelty is that, rather than using unstructured random bases as in NoLA, SOLAR uses structured, model-informed bases: it samples (r)-dimensional subspaces from the pretrained weight matrix’s SVD (left/right singular directions) and perturbs them with controlled random noise.

**Reviewer Concerns:**

The reviewers brought up a few concerns, for most of which the authors provided responses. For instance,

* Missing or insufficiently discussed simple post-hoc baselines. Multiple reviewers asked for a comparison against straightforward post-training compression baselines (notably SVD truncation / “drop singular values”) to better isolate what SOLAR contributes beyond “compress the adapter after training.” The authors provided PhLoRA and SVD truncation in rebuttal.

* Comparison with the quantization-based method! However, as correctly pointed out by the authors, SOLAR is complementary/orthogonal to quantization and can further push compression using quantization.

* A core assumption in this paper is that $\Delta W$ can be well-expressed in a subspace aligned with the base model. This is not trivial/obvious, and must be stress tested. The authors mentioned that they add noise and that can handle the non-aligned case, which weakens their argument in the first place for selecting these subspaces.

* The most negative review argued parts of the evaluation are not representative (e.g., focusing on a 10-sample fine-tuning regime), and that reliance on older/smaller benchmarks or models (e.g., GPT-2, CIFAR-10/100) weakens relevance/impact, even if some larger-scale models are also included.

**Reviewer Scores:**

The paper received two reviews that are marginally above the acceptance threshold, one that is marginally below, and one reject, making the overall decision genuinely borderline.

After reading the paper carefully, I believe there may be a misunderstanding in how the authors characterize and compare against **NoLA**, which is the closest head-to-head baseline. In NoLA, the LoRA factors are expressed as linear combinations of random basis matrices, e.g.,

$$
\tilde{A}=\sum_{i=1}^{K}\alpha_i U_i \quad \text{and} \quad \tilde{B}=\sum_{j=1}^{K}\beta_j V_j,
$$

where $U_i$ and $V_j$ are random matrices, and the resulting update is

$$
\tilde{\Delta W}=\tilde{A}\tilde{B}.
$$

Hence, NoLA also induces a low-rank update—just like LoRA and the proposed SOLAR—and has an associated effective rank/budget. However, in the reported results (e.g., Table 1), rank is reported for LoRA and SOLAR but not for NoLA. In addition, the paper describes NoLA as using “unstructured” random matrices for $\Delta W$, which does not seem to be the right framing if NoLA is reparameterizing the LoRA factors rather than directly parameterizing $\Delta W$. This creates ambiguity about whether the comparison is truly like-for-like and raises concerns about the fairness of the evaluation against the most relevant baseline.

For these reasons, I lean toward rejecting the paper.

---

### Decision · Program_Chairs · 2026-01-26

Reject